# CHEX-seq detects single-cell genomic single-stranded DNA with catalytical potential

Youtao Lu [1,6], Jaehee Lee [2,6], Jifen Li [2], Srinivasa Rao Allu [3], Jinhui Wang[2], HyunBum Kim[2], Kevin L. Bullaughey[1], Stephen A. Fisher[1], C. Erik Nordgren[1], Jean G. Rosario[1], Stewart A. Anderson[4], Alexandra V. Ulyanova [5], Steven Brem[5], H. Isaac Chen[5], John A. Wolf[5], M. Sean Grady[5], Sergei A. Vinogradov[3], Junhyong Kim [1,7] & James Eberwine[2,7] ✉

Genomic DNA (gDNA) undergoes structural interconversion between single- and double-stranded states during transcription, DNA repair and replication, which is critical for cellular homeostasis. We describe "CHEX-seq" which identifies the single-stranded DNA (ssDNA) in situ in individual cells. CHEX-seq uses 3'-terminal blocked, light-activatable probes to prime the copying of ssDNA into complementary DNA that is sequenced, thereby reporting the genome-wide single-stranded chromatin landscape. CHEX-seq is benchmarked in human K562 cells, and its utilities are demonstrated in cultures of mouse and human brain cells as well as immunostained spatially localized neurons in brain sections. The amount of ssDNA is dynamically regulated in response to perturbation. CHEX-seq also identifies single-stranded regions of mitochondrial DNA in single cells. Surprisingly, CHEX-seq identifies single-stranded loci in mouse and human gDNA that catalyze porphyrin metalation in vitro, suggesting a catalytic activity for genomic ssDNA. We posit that endogenous DNA enzymatic activity is a function of genomic ssDNA.

DNA 3D structure of eukaryotic cells is well recognized for regulating chromosome organization and RNA transcription. Apart from the B-form, double-stranded conformation[1], DNA also takes the non-B, single-stranded form[2]. Stable ssDNA is thought to be involved in nucleosome localization, while transient ssDNA in replication, repair, recombination, and transcription. An example of transcriptionally related ssDNA is the transcription bubble[3,4]. Specifically, the bubble moves along the length of the gene being transcribed, and in concert, long stretches of single-stranded areas longer than a kilobase form over the transcriptionally active chromatin[1,3,5]. Single-stranded DNA exists not only in genic regions but also in intergenic regions, including sites of DNA repair and replication, and these ssDNAs can be short or long lived and are reported to be regulated by a family of ssDNA binding proteins[6]. The amount of ssDNA in the genome is estimated to vary from ~0.2% to 2.5%, depending upon the physiological state of the cell[5].

Study of transcription in single cells in their natural microenvironment has been difficult, not only in the small amount of input material per cell, but also in the need to analyze the chromatin status before it is biochemically isolated. Many chromatin studies have relied upon pooled cells to generate enough gDNA/chromatin for analysis. An partial list of such techniques includes FAIRE-seq[7], ChIP-seq[8], DNase-seq[9], and ATAC-seq[10]. Recently, these methods have been extended to single cells, for example, single-cell ATAC-seq by combinatorial cellular

[1]Department of Biology, School of Arts and Sciences, University of Pennsylvania, Philadelphia, PA 19104, USA. [2]Department of Systems Pharmacology and Translational Therapeutics Perelman School of Medicine, University of Pennsylvania, Philadelphia, PA 19104, USA. [3]Department of Biochemistry and Biophysics, Perelman School of Medicine, University of Pennsylvania, Philadelphia, PA 19104, USA. [4]Department of Psychiatry, Children's Hospital of Philadelphia, ARC 517, 3615 Civic Center Blvd, Philadelphia, PA 19104, USA. [5]Department of Neurosurgery, Perelman School of Medicine, University of Pennsylvania, Philadelphia, PA 19104, USA. [6]These authors contributed equally: Youtao Lu, Jaehee Lee. [7]These authors jointly supervised this work: Junhyong Kim, James Eberwine. ✉e-mail: eberwine@pennmedicine.upenn.edu

indexing[11] or by fluidic sorting[12], single-cell DNase-seq[13], single-cell ChIP-seq[14], and ATAC-see[15]. Each of these procedures detects open chromatin in isolated single nuclei, but to the best of our knowledge none except ATAC-see has been adapted for in situ localized single cells, which is yet limited by the difficulty in accessing double-stranded DNA (dsDNA) in cross-linked nuclei of fixed cells.

In our efforts to understand brain cells' functional dynamics (especially the transcriptional potential) and to complement the other single-cell chromatin approaches, we present a method named CHEX-seq (**CH**romatin **EX**posed). It assumes that, as regions of transcribed open chromatin exist in single-stranded form, if we chemically cross-link the chromatin and preserve the cytoarchitecture, then we would be able to detect and evaluate the single-stranded regions in their natural context. In this article we report the application of CHEX-seq to exploring ssDNA open chromatin in fixed dispersed neurons and astrocytes, and in situ localized single neurons preserving the cellular microenvironment.

## Results

### Overview of CHEX-seq Analysis and Experiment Design

To assay ssDNA at single-cell resolution in situ, we designed a ssDNA chromatin interrogator as a multifunctional oligonucleotide probe. It is composed of three parts: the 5' barcode, the degenerate sequence, and the 3' lightning terminator (Fig. 1a, Supplementary Fig. 1). The barcode can distinguish multiplexed samples from the same library. The degenerate sequence is designed to anneal to single-stranded gDNA in fixed cells (Supplementary Fig. 1, Supplementary Table 1). The lightning terminator is a fluorescently tagged, photo-reversibly blocked deoxynucleotide. It will remain inactive at the site of annealing until light activation, which liberates a free 3'-OH that serves as a primer for spatially-localized, polymerase-mediated in situ cDNA synthesis[16,17] (Fig. 1a). After in situ DNA synthesis, the cDNA was removed with 0.1 N NaOH, copied into dsDNA, and linearly amplified using the T7 RNA polymerase (a.k.a. aRNA amplification)[18,19]. The aRNA product was subsequently reverse transcribed to 1st and 2nd strand DNA with custom primers, PCR amplified, made into libraries, and sequenced using 75 bp paired-end reads (Supplementary Fig. 2). Raw reads were filtered based on the barcode/primer quality (Supplementary Fig. 3, Supplementary Data 2), aligned to the respective (human or mouse) reference genome, further filtered by the alignment quality (Supplementary Data 3), and finally the 5' end of the barcode-carrying reads was taken as the ssDNA site. It is noteworthy that, unlike other open-chromatin assays (ATAC-, DNase-, FAIRE-seq), the barcoded design and strand-specific prep enable CHEX-seq to preserve ssDNA's strand information (For details see Methods).

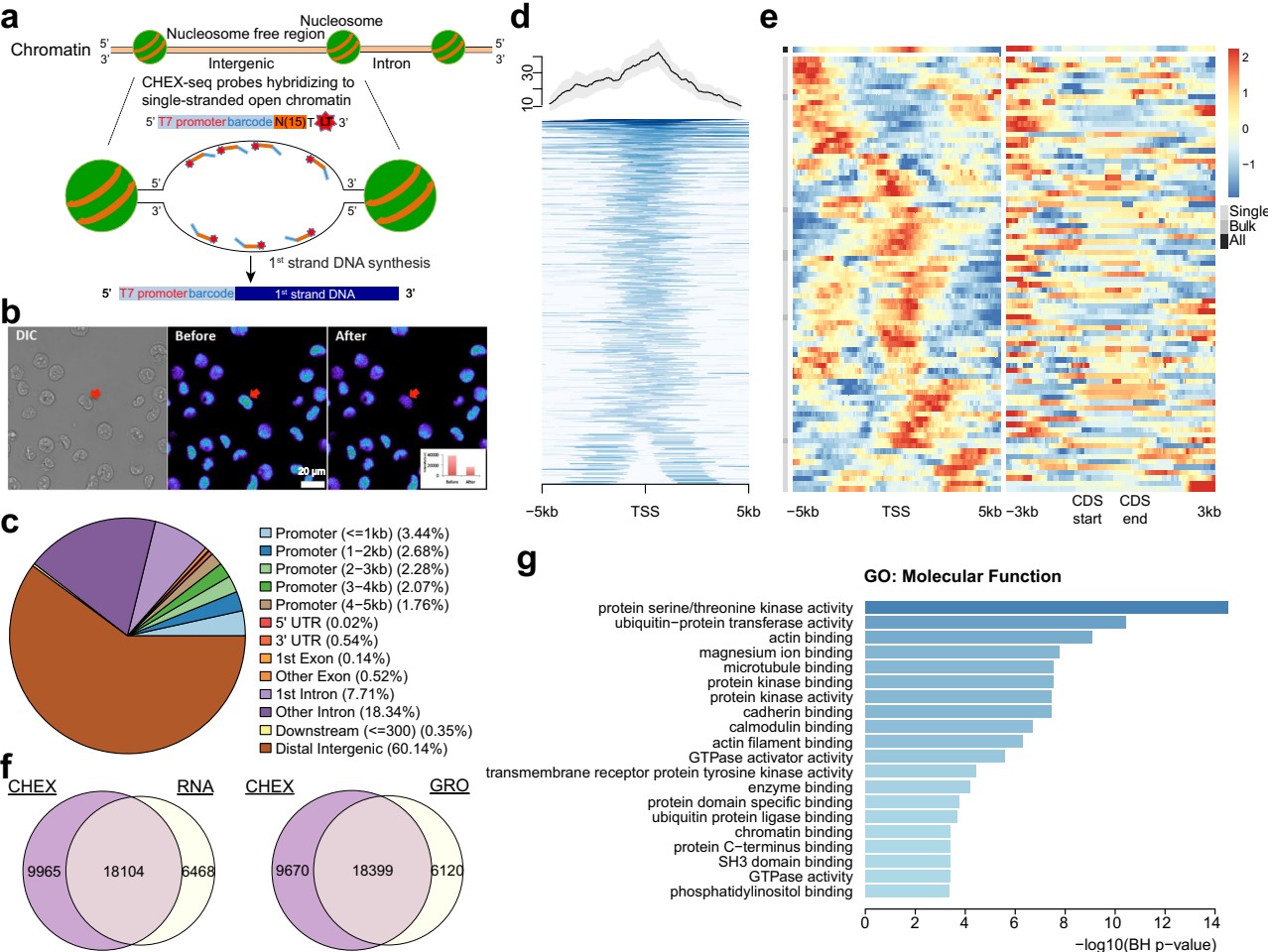

**Fig. 1 | CHEX-seq experimental design, K562 priming location characterization, and overlap with K562 transcriptome. a** Schematic of CHEX-seq assay; **b** Photoactivation at a specific site in the nucleus of K562; inset shows reduction in fluorescence upon photoactivation for a single nucleus but has been done every time the CHEXseq probe is activated in a single dispersed cell, *n* > 100. **c** Statistics of CHEX-seq priming sites with respect to genomic regions; **d** TSS proximal (+/− 5 kb) coverage of K562 samples (all positive non-outlier samples aggregated, top 1%-tile genes shown); the shade presents the mean ± SEM; **e** Z-scored coverage at TSS (+/−5kb) and CDS (+/−3kb) proximity at single-cell level; Single: single-cell samples, Bulk: multi-cell samples, All: aggregates of Single and Bulk; **f** Overlap between CHEX-seq primed genes (extended gene body > 0) and RNA-seq highly expressed genes (counts > median); **g** GO functional enrichment results (top 20 significant terms) of the CHEX-RNA overlapping genes (**f**, left), x-axis, -log10 of the *p*-value from hypergeometric test after Benjamini-Hochberg correction.

To show the utility of CHEX-seq, we ran extensive tests on distinct species, cell types as well as experimental conditions. They include two species (human and mouse), two classes of brain cells (astrocytes and neurons), two types of neuronal cell preparations (dispersed primary culture and in situ tissue section), and finally, K562 cells perturbed by TPA (12-O-Tetradecanoylphorbol-13-Acetate), a protein kinase C (PKC) activator and peripheral blood lymphocyte mitogen[20,21], which can induce chromatin accessibility changes in K562. The wide range of applications proves a general utility of our methodology, and permits interesting biology correlating ssDNA to genome maintenance, replication, and transcriptional activity to be uncovered.

## CHEX-seq Benchmark in Human K562 cells

Human K562 cells were chosen by ENCODE for extensive chromatin analyses[22,23]; hence we selected this cell line for benchmark. After fixation, K562 cells were gravity deposited onto poly-L-lysine-coated cover slips, then permeabilized and washed in PBS. Annealing of the CHEX-seq probes to the fixed cells shows the tagged fluorescence concentrating in the nucleus of the cell (Fig. 1b, middle). The lightning terminator was illuminated by 405 nm (UV) laser at 60% power and 30μs per pixel, whereupon a 45‑80% decrease in the fluorescence was observed (Fig. 1b, right). This indicates the loss of the fluorescent moiety and the freeing of a 3'-hydroxyl group that can later prime DNA synthesis.

After a series of QC filtering (for details see Methods: Data Preprocessing), the total number of priming sites in each non-control K562 sample varies from 305 to 60,437 (median = 2640) for single cells, and from 30,118 to 85,382 (median = 53,357) for multi-cell (bulk) samples (Supplementary Fig. 4). As a type of negative control, we treated 14 K562 cells with mung bean nuclease to digest ssDNA, and we observed significant reduction in the total number of priming sites, ranging from 1,139 to 4632 (median = 1890). The background is due to mung bean nuclease's incomplete digestion of ssDNA. Further, we carried out two other types of control experiment: (1) with barcoded probes but no laser activation ("Probe(+) Laser(-)"); (2) with neither barcoded probes nor laser activation ("Probe(-) Laser(-)"). Probe(-) Laser(-) controls showed greater reduction: only a median of 381 or 14,704 total priming sites observed in single cells or bulk samples, respectively (Supplementary Fig. 4a, left), suggesting an overall false positive rate of 14.4% and 27.6% in single or bulk samples, which is higher than the expected 6.7% (1/16 for the 2bp-clipped primer induced endogenous priming). Several control samples were found to have reads of low sequence complexity due to alignment artefacts or are linked to a specific probe (517 s) which could have problematic annealing or extension chemistry (for details see Methods: Background Estimation and Processing). For other controls, we noted that C reads, i.e. reads with lesser barcode/primer quality (for definition see Methods: CHEX-seq ssDNA Calls and Priming Counts) were more prevalent in controls than non-controls (Supplementary Fig. 4a, right), indicating that the background in these controls are spurious. Because CHEX-seq priming was sparse particularly in single cells, to ensure that we report as much of the complexity ssDNA regions as possible we prioritized the sensitivity and conducted a detailed examination of the background reads and showed that the impact of the background in non-control samples is negligible (for details see Methods: Background Estimation and Processing). Collectively, we conclude that CHEX-seq has a meaningful specificity in ssDNA detection despite a relatively high background in bulk controls.

We computed the percentage of CHEX-seq priming sites that map to the gene body, the flanking regions, including the 5' promoter, TSS, exons, introns, the 3'-proximal area, and distal intergenic regions (Fig. 1c). K562 cells show the highest proportion of priming sites in intergenic regions (> 60.1%), followed by introns (~26.1%), and then by promoter regions (less than 5 kb to the TSS) (~12.2%). Further breaking down the TSS 5 kb neighborhood into the proximal (<1 kb) and the distal (4–5 kb) regions, we observed almost two times more priming

sites in proximal regions than distal regions (1.8%), consistent with the notion that chromatin tends to be more accessible near the TSS. Indeed, we observed TSS enrichment in most non-control samples, while weak or no enrichment in negative controls (Supplementary Fig. 4c–e). Combining the coverage across all non-control samples created a distinct peak centered at the TSS (Fig. 1d), resembling the TSS peak observed in ATAC-seq[10]. However, closer comparison revealed distinction between the two assays: ATAC-seq has a sharp peak around the TSS, while the CHEX-seq has a wider peak with an extended slope 5' to the TSS (Supplementary Fig. 5a). These findings suggest that there is an anticipatory single-strandedness to the TSS of genes that are likely to be transcribed[6]. Alternatively, these CHEX-seq sites found 5' of the TSS may result from TSS-proximal DNA being uncoiled as a result of the "bursting transcriptional activity"[24] at the TSS. Figure 1e shows CHEX-seq coverage from single-cell, bulk, and sample aggregates, pooling annotated features (gene or coding sequence [CDS]): the left panel shows cell-cell variability in CHEX-seq priming locations near the TSS, and the right panel shows an increased propensity for ssDNA near the start of the CDS.

To assess how many of the K562 priming sites correspond to expressed mRNA, we compared the CHEX-seq data with published K562 transcriptome datasets (Fig. 1f). About 73.6% (18,104) of the highly expressed (>median) genes measured by RNA-seq had corresponding CHEX-seq priming. Even with this relatively large overlap, there were still 35.5% (9,965) single-stranded genes that were not highly expressed. Further, we reasoned that transcriptionally associated ssDNA might be more correlated to ongoing transcription and nascent mRNAs, which could be detected by GRO-seq, a real-time transcription runoff assay[25]. Therefore, we compared CHEX-seq to GRO-seq transcripts[26] and observed similar but more overlapping genes (~75.0%) with a slight decrease in the CHEX-seq unique genes (~34.5%). Gene Ontology (GO)[27] enrichment analysis of the CHEX-RNA overlapping genes identified cell signaling, kinase activity, and GTPase regulatory pathways as significantly enriched, consistent with the fact that K562 cells are a transformed cell line (Fig. 1g).

## CHEX-seq compared to other open-chromatin assays

Having assessed CHEX-seq's TSS propensity and transcriptional association, we further benchmarked CHEX-seq against three other well established chromatin assays (ATAC-, DNase-, FAIRE-seq) in genome-wide coverage. Figure 2a exemplifies the comparison in an 800 kb region of Chromosome X (chrX:48,500,000-49,300,000) from the UCSC Genome Browser[28,29]. There is commonality between CHEX-seq and the other three methods, however, we note regions unique to each method. For example, the first dashed-line box highlights the gene OTUD5, whose 3' UTR and downstream area are shared by ATAC-seq and CHEX-seq, while the gene PIM2's intron is shared by CHEX-seq, DNase-seq and FAIRE-seq; the second dashed-line box highlights the sixth intron of CCDC22 where all the four methods overlap, though CHEX-seq appears slightly downstream the other three (Fig. 2a). We reason that the discrepancy can be explained by the fact that different epigenomic assays have different genomic scales due to both the biological nature of the signals detected by each technique and the chemistry of each assay.

To better quantify the genome-wide relationship between different open-chromatin assays, we computed signal concordance between CHEX-, ATAC-, DNase- and FAIRE-seq, against a select set of K562 epigenomes (broad and narrow histone modifications, DNA methylome, Pol II ChIP-seq, GRO-seq, super-enhancers, and replication origins) in fixed-sized bins followed by hierarchical clustering (Fig. 2b). At the size scale of 5 kb windows, the four chromatin methods cluster together while CHEX-seq appears the most distant (Fig. 2b, left). The GRO-seq transcriptome (in bold) is even more distant, indicating that there is not a one-to-one overlap between detectable open chromatin and newly transcribed genes, even with the pronounced overlap

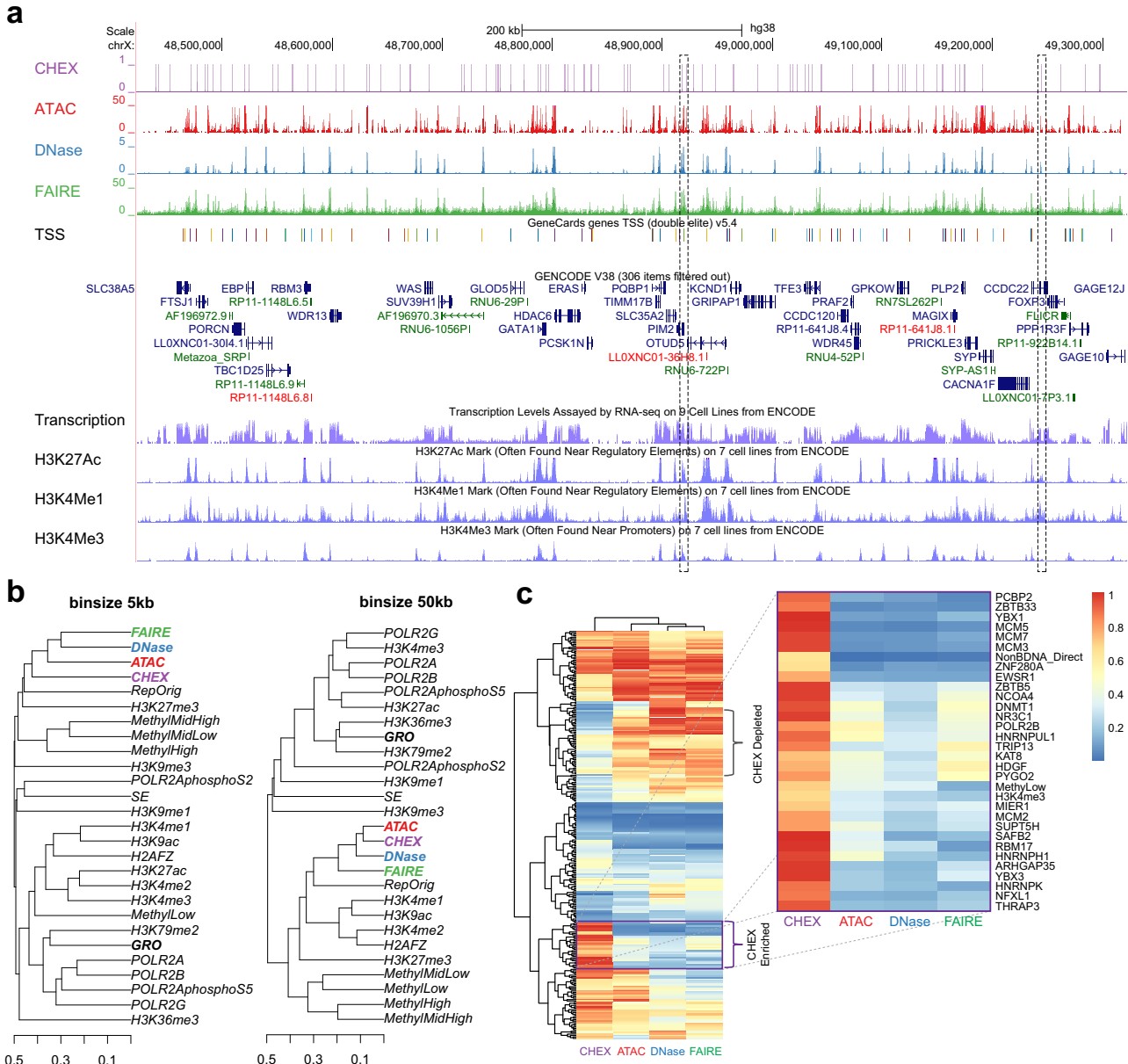

**Fig. 2 | Genomic comparison of CHEX-seq with other open-chromatin assays (ATAC-, DNase-, FAIRE-seq), transcriptome (GRO-seq), and epigenomes (RRBS DNA methylation, histone modifications, super enhancer [SE]). a** UCSC Genome Browser track view comparing the coverage of CHEX- (purple) against ATAC- (red), DNase- (blue) and FAIRE-seq (green) at locus OTUD5. Below four assays is the GeneCards TSS track. The last four tracks are transcriptome and three histone marks. Dashed-line boxes highlight two loci shared by all four open-chromatin assays; **b** Hierarchical clustering of open-chromatin assays, transcriptome, and epigenomes at 5 kb (left) or 50 kb (right) resolution, using binarized coverage and Jaccard distance; **c** Hierarchical clustering of CHEX-, ATAC-, DNase- and FAIRE-seq by the similarity with an extended set of 284 K562 epigenomes. Color indicates quantile normalized fold of enrichment given a particular assay (0 means the lowest enrichment and 1 means the highest enrichment).

between CHEX-seq and highly transcribed nascent mRNAs as shown in Fig. 1f. At the size scale of 50 kb, again, FAIRE-, ATAC- and DNase-seq form a cluster with CHEX-seq, with CHEX-seq displaying the closest relationship to ATAC-seq (Fig. 2B, right). As the average size of a human gene is ~42 kb and the functional transcriptional chromatin unit is ~50 kb[30], these data suggest that many of the same open-chromatin associated genes are identified with each of these procedures, but the ssDNA identified by CHEX-seq is distinct from the dsDNA open chromatin identified by the other approaches. A direct overlap would not be expected, as the other procedures have a target bias for dsDNA (ATAC- and FAIRE-seq) or are indiscriminate of single or double strands (DNase-seq) as compared to CHEX-seq's ssDNA specificity. Moreover, CHEX-seq data are sparser due to fewer numbers of

analyzed cells. These results highlight that both double-stranded and single-stranded DNA exist within the open chromatin of genic and intergenic genomic areas, both of which sculpt the open-chromatin landscape of a cell.

In addition to the clustering analysis comparing CHEX-seq with the select set of K562 epigenomes (Fig. 2a, b), we extended this analysis to a full set of 284 K562 epigenomes (for a complete list see Supplementary Data 4) curated from ENCODE ChIP-seq[31], non-B-form DNA database[32] and R-loop DRIP-seq[33], and asked how CHEX-seq differs from other three open-chromatin assays in the epigenome. We calculated genomic association scores[34] for each of the four chromatin assays to each epigenome (Supplementary Data 4), then normalized the enrichment scores to rank quantiles to alleviate assay-specific

confounding factors (such as different coverages or peak breadths), and finally clustered the four assays based on the normalized enrichment scores (Fig. 2c). As highlighted in the zoom-in view of Fig. 2C, CHEX-seq exhibits strong overlap with a unique set of transcription factors (TFs), including the Y-box family (YBX1/3), the minichromosome maintenance family (MCM2/3/5/7), non-B-form DNA (direct repeats), and heterogeneous nuclear ribonucleoproteins [hnRNPs] (HNRNPH1, HNRNPUL1, HNRNPK). The Y-box and the MCM protein family have been well known as evolutionarily conserved ssDNA binding proteins that participate in DNA replication and repair[35,36]. Interestingly, both protein families are related to transcription as well. For example, YBX1 is reported for the involvement in premRNA transcription, splicing, mRNA packaging and stability regulation[37], and similarly, MCM2–7 for being necessary for RNA Pol II-mediated transcription[38]. Some members of the hnRNPs are known to bind to single-stranded telomeric DNA in vitro[39], hence we are less surprised by the enrichment in hnRNPs. In fact, TFs with DNA/RNA dual binding capacities have been shown previously (Cassiday and Maher III 2002).

More intriguingly, we found that DNMT1, one DNA methyltransferase that is responsible for conferring the template strand's methylated status to the newly synthesized strand during replication[40], ranked much higher for its enrichment in CHEX-seq (18th of 284) than ATAC- (152nd of 284), DNase- (221st of 284) or FARE-seq (147th of 284) (Fig. 2c). Although CHEX showed higher enrichment in low-/unmethylated genome (MethyLow) than other three assays (Fig. 2c), it suggests a possible role of ssDNA in DNA methylome maintenance, which is also a single-stranded process. Besides the enrichments, we also noticed some epigenomes are exclusively depleted in CHEX-seq. They are mostly TFs, and many participate in development, proliferation, or neoplasia, e.g., GATA1, JUNB, JUND, MYC, etc. (Fig. 2c, upper brace). These sites could represent genes whose expression is limited to a particular stage of development or disease state (oncogenesis) which is reflected in their single-strand/double-strand ratios.

To reduce spurious reads, we tightened the barcode/primer class criterion (quality level from A/B/C to A/B only) and the alignment criterion (minimal mapped length from 20 bp to 30 bp), then repeated this analysis. This time, we not only recapitulated the enrichments (YBX1/3, MCM3/5/7, HNRNPK, DNMT1), but also uncovered additional CHEX-seq exclusive epigenomes: R-loops, super-enhancers, and transcriptional activity indicators (POLR2AphosphoS2, GRO-seq), and other types of non-B DNA (short tandem and mirror repeats) (Supplementary Fig. 6a). In summary, the benchmark results showed ample evidence for CHEX-seq's ability to detect ssDNA in the process of expression, replication and possibly DNA methylation maintenance.

## FISH validation of CHEX-seq identified intergenic ssDNA Loci

To validate the CHEX-seq predicted ssDNA loci, we performed single-molecule FISH (smFISH) for a CHEX-seq priming hotspot on Chromosome 1 (chr1:630737–633960), where ATAC-seq predicted open while DNase-seq predicted limited openness and FAIRE-seq predicted closed (Supplementary Fig. 7a). Eight 20-mer oligonucleotide probes were synthesized to target this area and these probes were labeled at the 5′-end with the ATTO 590 fluorophore. It is important to note that, as we were assessing endogenous single-strandedness, no heat denaturation of the tissue was performed. Therefore, a positive signal could only arise from endogenous single-stranded chromatin. Three strong positive spots are observed in a single nucleus with smFISH (Supplementary Fig. 7b). This trisomy signal is due to the complicated K562 cell karyotype where some cells have 3 copies of Chromosome 1[41]. To further test CHEX-seq and to rule out the interference of RNA transcripts during probe hybridization, we did a second run of smFISH experiments on other four additional loci from distal intergenic regions of Chromosome 4, 8, 11 and X (Supplementary Data 5). All showed at least one positive cell out of ~10 cells in the field of view

(Supplementary Fig. 7c–f). Specifically, loci chr4:1466001–1468000 (Supplementary Fig. 7c) and chr11:123002001–123004000 (Supplementary Fig. 7e) showed ~30% cells with two or more fluorescent spots. In conclusion, our FISH experiment corroborated that single-stranded chromatin structure is detected by CHEX-seq.

## The Role of ssDNA in Active Transcription and the CHEX-seq Strand Specific Model

Having established CHEX-seq as a ssDNA detection method, we went on to explore how single-stranded chromatin is associated with transcription in a more quantitative way. We stratified genes in K562 according to their CHEX-seq priming location (if any) with respect to the distance to the TSS, then correlated it to the same gene's mRNA expression level from three different sources – bulk RNA-seq[42], bulk GRO-seq[43], and single-cell RNA-seq [scRNA-seq][44]. We observed an anticorrelation between the CHEX-seq priming distance to the TSS and the gene expression level for all three datasets: the closer the CHEX-seq priming sites to the TSS, the higher the median expression level of the corresponding mRNAs (Fig. 3a–c). The same pattern was also found in human and mouse brain cells (Supplementary Fig. 8a–c and d–g, respectively). These results suggest a regulated plasticity with regard to ssDNA within a gene: i.e., as the transcription machinery moves along the length of the gene, the 5′-open site becomes unavailable for hybridization, perhaps due to reannealing of the single-stranded region or formation of an intrastrand DNA secondary structure[45]. This suggests that CHEX-seq priming sites have varying half-lives, and those that are proximal to the TSS remain single-stranded for a longer time and correspond to high levels of transcription, thus CHEX-seq priming closer to the TSS is predictive of highly transcribed genes. We would like to note that these are not simply due to the differences in RNA stability, as GRO-seq detects newly synthesized nuclear RNA but shows the same anticorrelation (Fig. 3b). We hence postulate a location dependent model of ssDNA: TSS-proximal ssDNA is associated with ongoing gene expression whose single-stranded accessibility decays with 5′ to 3′ progress of transcription, while distal ssDNA is more likely related to other conformational regulation of the DNA such as genome replication or methylome maintenance.

Unlike other open-chromatin assays such as ATAC-, DNase- or FAIRE-seq, CHEX-seq identifies the DNA strand that is being copied as its primer-extension in situ transcription method informs sequence directionality. Since the RNA Pol II transcriptional complex binds to the DNA and synthesizes mRNA transcripts in a 5′ to 3′ direction by transcribing the antisense (template) strand, we hypothesized that CHEX-seq probes might be preferentially bound to the potentially more accessible sense strand, giving rise to an excess of antisense-strand "CHEX-seq transcripts" (Fig. 3d). We calculated the ratio of antisense to sense-strand reads for different regions of the annotated gene model: TSS 5 kb upstream, TSS 5 kb downstream, exons and introns. As expected, we found generally more antisense-stranded reads in the transcribed regions of the genome, and the strongest antisense bias was in the TSS 5 kb upstream, followed by the TSS 5 kb downstream, then by the exonic region (Fig. 3e). In contrast, the intronic region showed the least bias (Fig. 3e). We reason the strongest antisense bias in TSS proximity to be due to promoter-proximal RNA Pol II pausing as a pervasive transcription regulatory mechanism in metazoans[46,47]. We speculate the disappearance of the bias from exons to introns was a result of the uneven distribution of RNA Pol II along the gene. As observed in mouse embryotic stem cells (mESCs), RNA Pol II has peak density at the TSS, then rapid decay after the TSS, with a gradual rebound towards the termination site[48]. Therefore, the antisense strand of the first and the last exon are hypothesized to show greater RNA Pol II occupancy than other exons or introns. Herein, we established CHEX-seq's TSS-proximal signal as an indicator of active transcription, and further proposed and tested our model for CHEX-seq

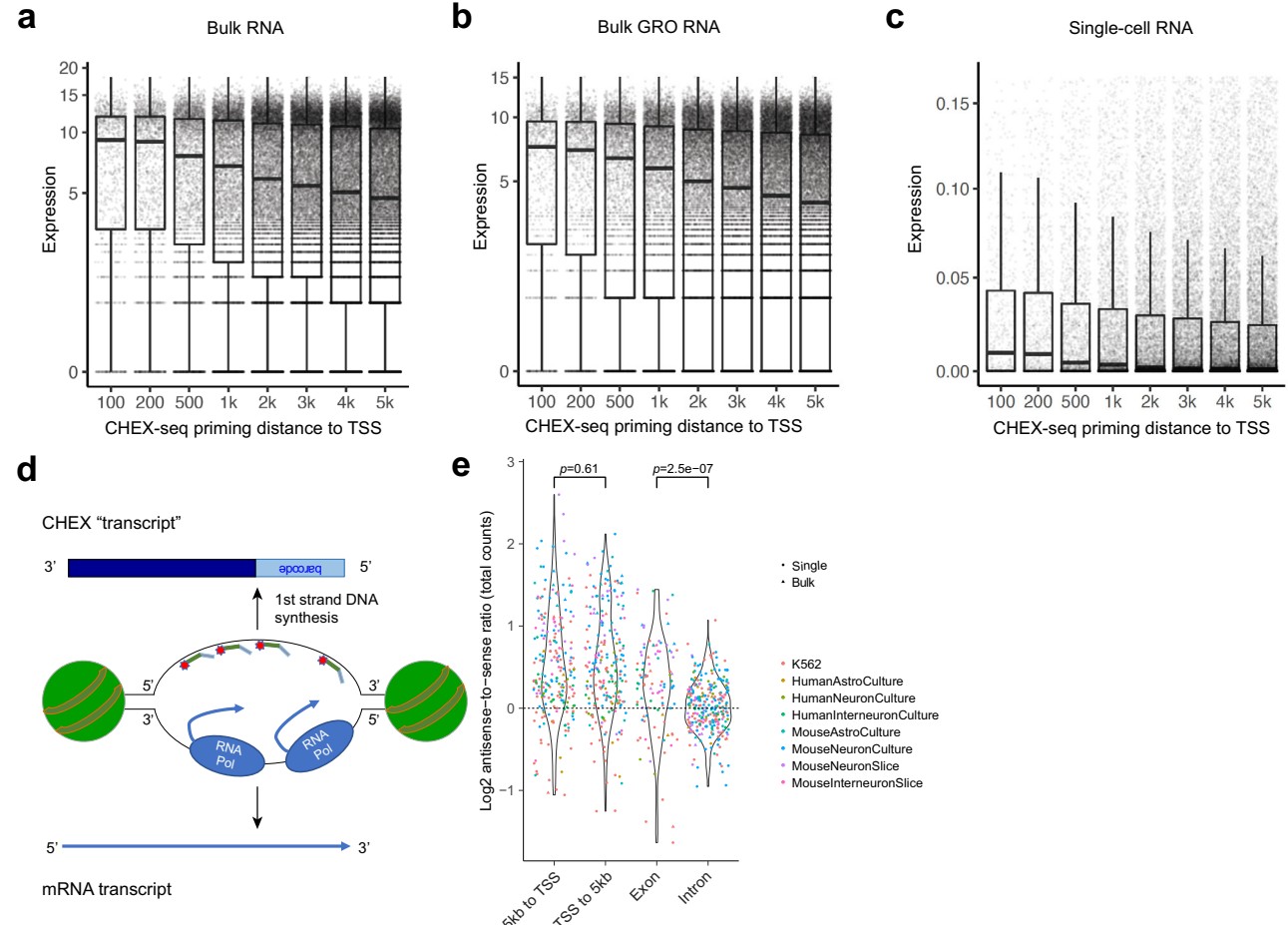

**Fig. 3 | Expression levels of genes with TSS priming stratified by the distance to TSS (maximum 2.5 kb to either direction). a** Bulk K562 RNA-seq; **b** Bulk K562 GRO-seq; **c** K562 scRNA-seq, single cells aggregated. Y-axis, gene expressions (capped at 90%-tile); x-axis, distance (bp) to TSS from CHEX-seq priming sites; $n = 81$ biologically independent samples. The bounds of the box represent the 1$^{st}$ and the 3$^{rd}$ quartile; the thick bar represents the median; the whiskers extend 1.5 times the interquartile range (IQR); the dots represent all data points including maxima and minima. **d**–**e** CHEX-seq unique property: detecting open chromatin's strandedness. **d** Schematic showing the hypothesis that CHEX-seq reads should have opposite strandedness than sense-strand mRNA transcripts; **e** Testing the hypothesis in (**d**). X-axis, sub-genic regions; y-axis, ratio of the number of antisense-stranded over sense-stranded CHEX-seq reads, samples with less than 5 counts discarded; significance by Wilcoxon rank-sum test (two-sided).

strand bias due to polymerase/TF occupancy preference in transcribed regions.

## Dynamic regulation of ssDNA upon chromatin perturbation in K562 cells

Given the ability of CHEX-seq to identify single-stranded open chromatin, we wondered if these regions are dynamically regulated. To investigate this, we treated K562 cells with TPA, a PKC activator known to modulate K562 gene expression and induce differentiation in leukemic cells[49–51]. To capture the time-dependent changes in ssDNA, we aggregated the single-cell replicates at each time point: pre-treatment, 15 min, 1 h, 2- and 24-hours post-treatment, and plotted the cumulative distribution of ssDNA as a function of the priming distance to the TSS, which was further summarized as the fold of enrichment score at each time point (Fig. 4a). As anticipated, we observed the highest fold of ssDNA enrichment around the TSS in untreated samples, and upon treatment, a steep decrease at 15 min after treatment, followed by continued decrease leading to the minimum at 1 h. After that, the TSS ssDNA enrichment began to recover, and reached increasingly higher level at 2 and 24 h, though the end-point enrichment still fell short of the level at 15 min. Collectively, this time-dependent trend suggested a rapid and time-dependent regulation of single-stranded open chromatin in K562 cells.

Given that TSS ssDNA enrichment showed the sharpest decrease at 15 min, we asked what genes or pathways perturbed ssDNA openness compared to the pre-treatment baseline. As shown in Fig. 4b, ssDNA regions in 67 genes were found to be highly differentially regulated immediately upon perturbation. Among them 6 genes (ADCY2, RBFOX1, TIAM2, NDUAF1, ZNF536, ZFAT) showed increased ssDNA openness, 61 genes exhibited decreased ssDNA openness. GO enrichment analysis suggested that the genes gained ssDNA openness are enriched in *cellular response to forskolin* (GO:1904322), *cAMP biosynthetic process* (GO:0006171) and *activation of protein kinase A activity* (GO:0034199) at an adjusted *p*-value 0.03. The genes that lost ssDNA openness are only significantly (adjusted *p*-value 0.04) enriched in *limb development* (GO:0060173). Despite lacking statistical significance, we still observed several TPA target-relevant genes change in ssDNA architecture (Fig. 4b, table inset). For instance, HRAT92 is a lincRNA specifically expressed in the heart[52], and CHD7 and MEGF8 are members of *heart morphogenesis* (GO:0003007). Consistent with our finding, TPA has been reported to be able to regulate cardiomyogenesis via PKC/ERK signaling in mESCs[53]. Moreover, MCTS1 and PDGFA are components of the *regulation of cell population proliferation* (GO:0042127), RAC3 is involved in *regulation of small GTPase-mediated signal transduction* (GO:0051056), and RASA4 is involved in *negative regulation of GTPase activity* (GO:0034260). These findings generally agree with

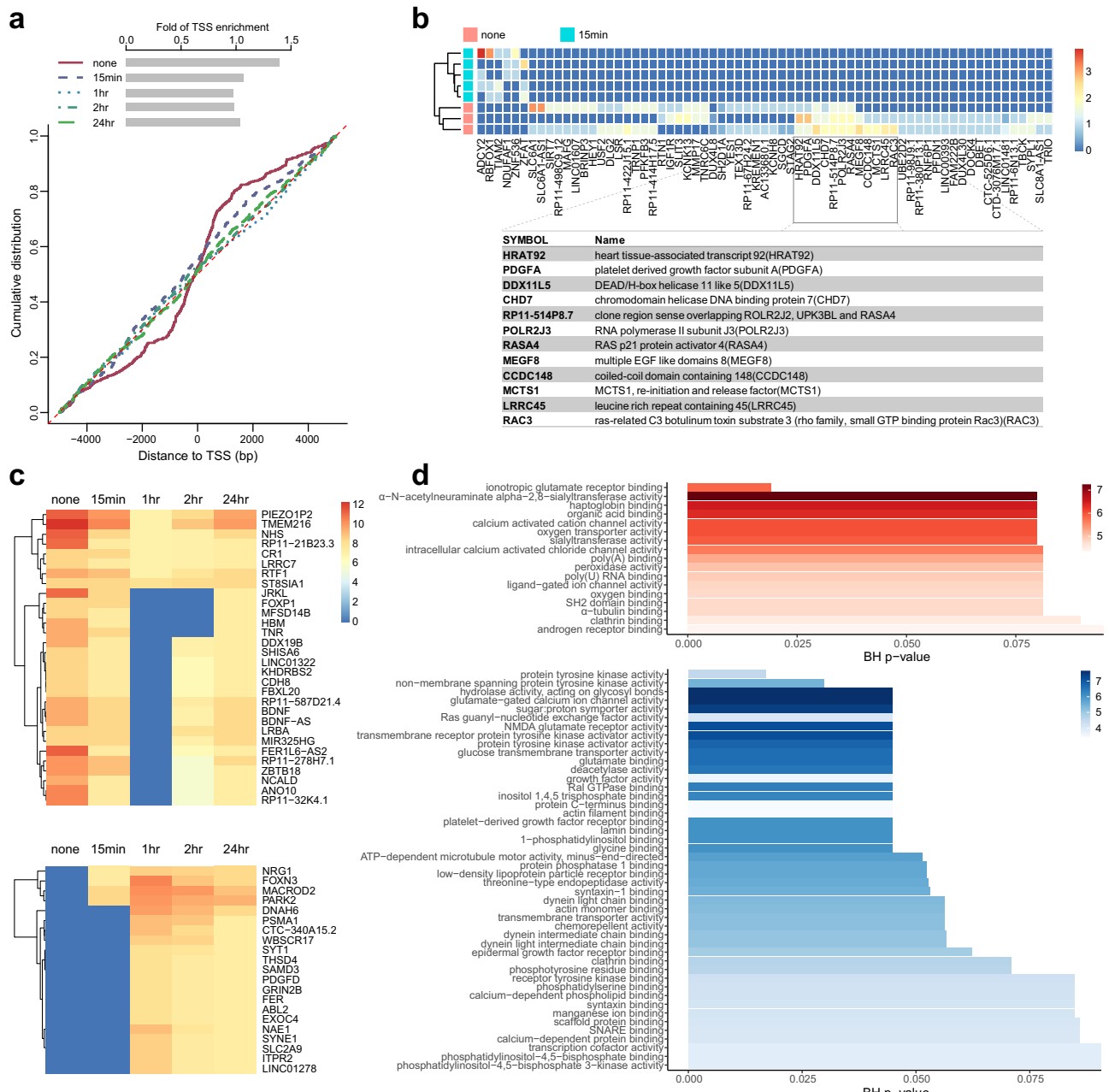

**Fig. 4 | TPA-induced dynamics in distribution of ssDNA throughout the genome. a** Changes in the distribution of CHEX-seq priming location (relative to TSS) upon TPA perturbation. Y-axis: cumulative empirical distribution of CHEX-seq priming sites with the distance to TSS; x-axis: distance to TSS (bp); **b** Differentially primed genes in acute response to TPA treatment. The heatmap color indicates the priming counts in log2; **c** Genes losing (top) or gaining (bottom) TSS single-strandedness in concert to the TSS enrichment changes. The heatmap color indicates the priming counts in log2. **d** Top 50 GO (Molecular Function) terms enriched in the genes losing TSS single-strandedness; significance by hypergeometric test.

TPA's mitogenic or pro-differentiation effect on blood cells[21,54] and highlight the ssDNA dynamics upon pharmacological treatment.

We further asked what genes had ssDNA openness covarying with the TSS enrichment trend across the five-time points. Our rationale was that given the correlation between CHEX-seq and active transcription, we would expect TPA-induced dynamics to be reflected in the nascent transcriptome, which in turn should correlate with CHEX-seq priming in the gene body. After filtering for genes whose CHEX-seq priming counts (library normalized) had high correlation ($|r| \geq 0.9$) with the TSS trend, we identified 30 genes that correlated with the TSS trend, and 21 genes that anticorrelated (Fig. 4c). These genes were queried for GO enrichment (Fig. 4d). LRR27 and SHISA6 from *ionotropic glutamate receptor binding* (GO:0035255) is the only significant function in the correlated

genes, while among the anticorrelated genes, several tyrosine kinase functions (GO:0004713, GO:0004715, GO:0030297, GO:0030296) and Ras (GO:0005088) and Ras-like (GO:0017160) functions are enriched (Fig. 4d, bottom). These results revealed the cell signaling and proliferation-related pathways show concerted regulation of TSS ssDNA chromatin dynamics. This demonstrates CHEX-seq's capacity to capture immediate and time-delayed changes in the single-stranded chromatin landscape upon chemical perturbation.

## CHEX-seq applied to spatially identified cells in mouse brain tissue sections

We investigated open-chromatin ssDNA landscape in individual neurons from in situ tissue sections of the adult mouse brain, where

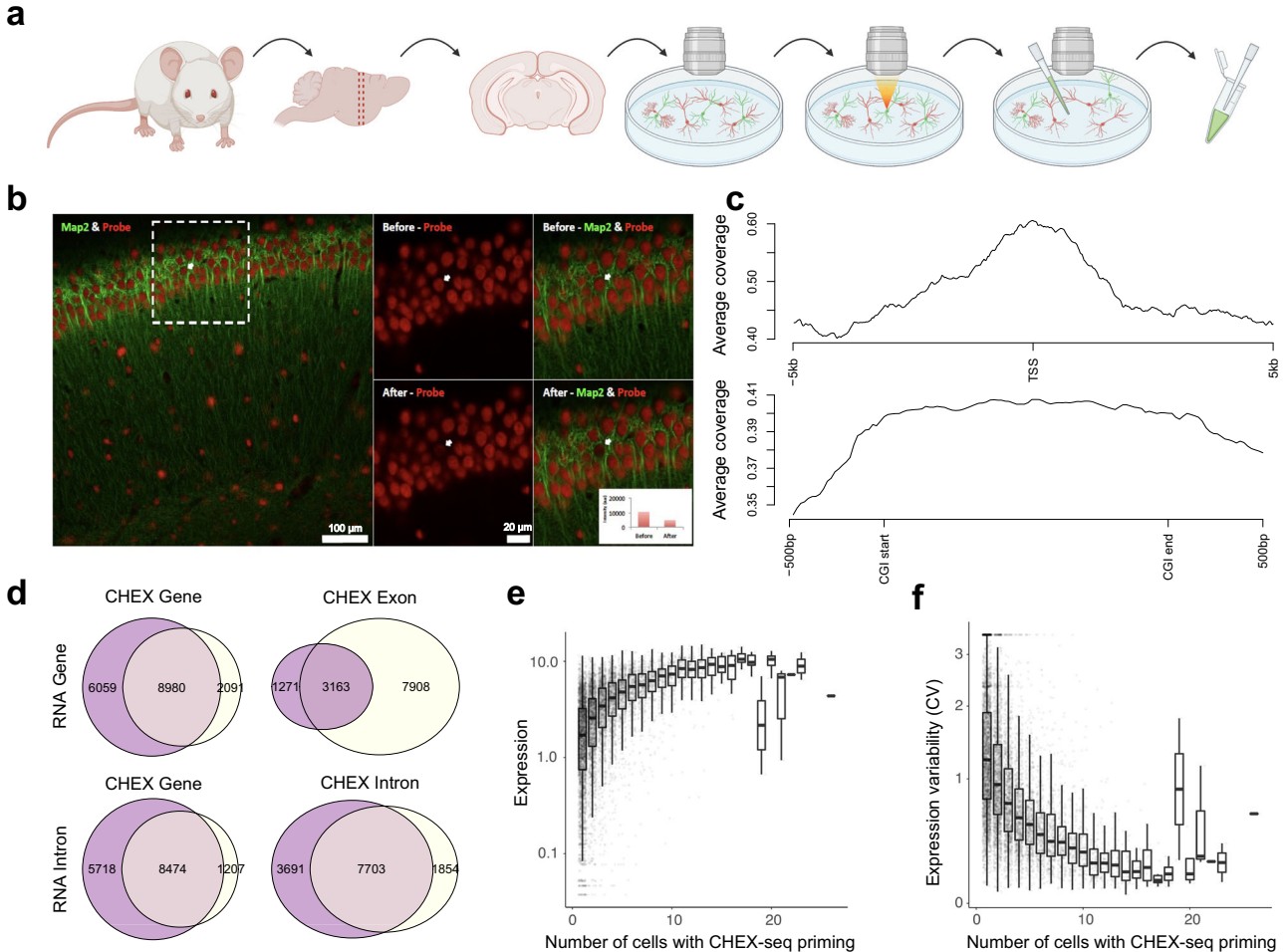

**Fig. 5 | CHEX-seq applied to mouse neuronal samples (fixed tissue section and dispersed culture). a** Schematic of CHEX-seq applied to mouse brain tissue section using cell-specific markers; **b** Photoactivation of CHEX-seq probes (red) at single neuronal cell stained with the marker *Map2* (green) in mouse brain section; inset shows reduction in fluorescence upon photoactivation for a single nucleus but has been done every time the CHEX-seq probe is activated on tissue sections, $n > 30$. **c** TSS ($\pm 5$ kb) (top) and CpG island flanking ($\pm 500$ bp) (bottom) CHEX-seq coverage in mouse brain section; **d** Overlap between CHEX-seq primed genes (>0) and RNA-seq highly expressed genes (counts > median). Rows are RNA-seq in whole gene body or intronic regions, columns are CHEX-seq in whole gene body, exonic or intronic regions. **e** Correlation between CHEX-seq intronic priming frequency and transcriptional activity (RNA-seq intronic counts) in mouse brain section, $n = 26$ biologically independent cells; **f** Correlation between CHEX-seq intronic priming frequency and transcriptional variability (RNA-seq intronic counts) in mouse brain section, $n = 26$ biologically independent cells. The bounds of the box represent the 1st and the 3rd quartile; the thick bar represents the median; the whiskers extend 1.5 times the IQR; the dots represent all data points including maxima and minima.

neurons were resident in their natural microenvironment (Fig. 5a). CHEX-seq probes were annealed to fixed adult mouse brain coronal tissue sections (100 μm) and detected by immunofluorescence using a fluorescently labeled antibody targeting the neuronal microtubule-associated protein 2 (MAP2)[55]. Figure 5b shows the CA1 region of the hippocampus stained with MAP2 immunofluorescence (green) and CHEX-seq probes (red). The CHEX-seq probes were then photoactivated in an individual nucleus (arrow in boxed area of Fig. 5b) with the activation confirmed by the loss of fluorescent signal (Fig. 5b, inset).

We first examined read coverage and confirmed the TSS peak when aggregating individual cells from multiple mouse neuron tissue sections (Fig. 5c, top). CpG islands (CGIs) also showed distinct elevation, especially in the area preceding the CGI start (Fig. 5c, bottom). Mammalian CGIs have been found to co-localize with promoters and replication origins[56–58], and these agree with ssDNA's role in transcription, replication, and putatively, DNA methylome maintenance as found in K562. We repeated CHEX-seq in mouse interneuron tissue sections and identified the same TSS peak (Supplementary Fig. 8c) and the CGI preference (Supplementary Fig. 8d). Additionally, we observed

in general higher coverage in the gene-body and a decay from the TSS to the TES (Transcription End Site) in mouse neuron (Supplementary Fig. 8a) and interneuron (Supplementary Fig. 8e) tissue section samples. A similar decaying pattern was also observed in the 5′ UTR region in these samples (Supplementary Fig. 9b, f), however, the functional implication of ssDNA in 5′ UTR is unclear.

Apart from the coverage, we compared the genes showing ssDNA from mouse neuron tissue sections with the high-expression transcriptome from single neuron RNA-seq. Like K562, we observed an overall significant overlap between CHEX-seq and transcriptome: ~60% of the ssDNA loci are highly expressed, while 81% of the highly expressed genes exhibit single-stranded regions (Fig. 5d, top-left). This asymmetry suggests a larger proportion of the single-stranded chromatin in fixed tissue sections that is not substantially represented in the transcribed mRNA, and they likely correspond to genes poised to transcribe, DNA replication sites or other types of DNA organizational structures. Further breaking down CHEX-seq or RNA-seq reads into exonic or intronic regions, we found an even stronger overlap with CHEX-seq and RNA-seq in introns (Fig. 5d, bottom-right). As discussed for the K562 CHEX-seq data, intronic transcripts are more indicative of

the active transcriptional events hence a better overlap in intronic RNA-seq is unsurprising (Fig. 5d, bottom-left). What was surprising was the better overlap in intronic ssDNA loci: ~68% of them are highly, actively transcribed, which indicates that in neuron tissue sections, single-stranded intronic chromatin has a better association with ongoing transcriptional activity. This may have to do with introns being longer than exons but may also be reflective of exons generally having a higher density of nucleosomes than exons[59].

Seeing the overall high CHEX-seq/RNA association in mouse neuron tissue sections, we hypothesized that a gene's ssDNA level would be positively correlated to the intronic expression level of that gene, and negatively correlated to the intronic expression variability of that gene. Using the number of single neurons that had CHEX-seq reads as a proxy for the ssDNA level, we confirmed both: monotonically greater expression level and monotonically lesser expression variability (CV) as the number of CHEX-seq positive neurons increased, especially when it did not exceed 18 cells (Fig. 5e, f). The genes that broke the monocity (on the right-most side) included Kif26b, Upk3bl, Ap3b1, Cadm2, Ctnna3, Gpc5, Mansc1, Ptprm, Tiam2, Mdga2, Tacc2 and Fgfr2; many of them are unusually long. For example, Fgfr2 has an isoform (Fgfr2–217) that contains an intron spanning ~3 Mb. Ctnna3 and Gpc5 have long isoforms (1.6 Mb and 1.4 Mb, respectively) too. We therefore suspect the weakened correlation was partly due to these extremely long genes. It is worth noting that, in general we did not find good correlation when using CHEX-seq read counts as the direct readout for the ssDNA level. Two reasons might explain this observation. First, we hypothesize that, for individual cells CHEX-seq might be more likely a binary gauge for ssDNA—it detects ssDNA by the presence or absence of priming, rather than the frequency of priming. Second, "bursting transcription" has been proposed to model the single-cell transcriptome[24,60]. A more highly expressed gene could trigger transcriptional bursts more frequently and the single-stranded state be present more often through the extended periods of bursting: thus, the number of priming cells (as an ensemble) would be a more representative statistic of the ssDNA level than the number of probes annealing to the ssDNA in a single cell.

## CHEX-seq replicated in human and mouse brain-dispersed cultures

Besides mouse in situ brain sections, we performed CHEX-seq analysis in human and mouse brain primary cultures and were able to recapitulate the relationship between CHEX-seq and transcription, non-B DNA and open-chromatin epigenomes. Supplementary Fig. 8 showed generally well-correlated, negative relationship between the RNA-seq expression level and the distance of the CHEX-seq priming sites to the TSS (within 5 kb) in human astrocyte (Supplementary Fig. 9a), neuron (Supplementary Fig. 9b), and interneuron (Supplementary Fig. 9c) cultures, and in mouse astrocyte (Supplementary Fig. 9d) and neuron (Supplementary Fig. 9e) cultures. The negative correlation was also found for both mouse neuron (Supplementary Fig. 9f) and interneuron (Supplementary Fig 8g) bearing tissue sections. The human and mouse brain samples recapitulated the ENCODE epigenomes enriched in K562. For human astrocyte and neuron cultures, we observed consistent and strong enrichment in non-B DNA (direct, short tandem, a-phased), but almost no enrichment in ATAC-seq or FAIRE-seq. This is possibly due to the ENCODE human brain chromatin data from control subjects did not match well with our subjects who presented in the clinic with epilepsy or brain tumors. Nonetheless, ENCODE human DNase-seq showed weak enrichment (~1.2X) in human astrocyte and neuron CHEX-seq (Supplementary Fig. 6b). For mouse astrocyte and neuron cultures, as well as neuron and interneuron tissue sections, we observed enrichment in non-B DNA (direct, mirror, short tandem, a-phased), ENCODE mouse brain H3K27ac, H3K27me3 and H3K4me3, though to a mild extent (~1.5X) (Supplementary Fig. 6c).

Having observed a tight association between CHEX-seq priming and RNA-seq expression in intronic regions in mouse neuron tissue sections, we asked whether other sub-genic regions also show such associations. To this end, we constructed genic units of GeneExt (the gene-body plus 2.5 kb upstream and downstream), Promoter (-2kb to +200 bp from the TSS), Exon, Intron and Downstream (from the TES to 2.5 kb downstream) (Supplementary Fig. 10a). For each (sub-)genic region, we pooled the priming counts from biological replicates and conducted Fisher's exact test for the association between CHEX-seq priming and the highly expressed genes from the RNA-seq of the corresponding cell type. We confirmed that the intronic genic ssDNA correlated best with the intronic gene expression in both K562 and human or mouse neuronal cell samples (Supplementary Fig. 10b). To further ask whether the CHEX-RNA association exists in single cells, we repeated the CHEX-RNA association test for individual K562 sample. Remarkably, we confirmed that genes with intronic ssDNA are significantly associated with highly expressed genes from K562 GRO-seq (Supplementary Fig. 10c), albeit to a lesser degree as compared to bulk or pooled CHEX-seq samples.

## CHEX-seq also detects mitochondrial ssDNA in single cells

As mitochondria also exist in fixed cells, we questioned if CHEX-seq could detect single-stranded DNA in the mitochondrial genome. Mitochondrial DNA (mtDNA) has been noted in other open-chromatin assays such as ATAC-seq, but has generally been removed in favor of nuclear DNA analysis[61]. Unlike nuclear DNA, mtDNA is not organized into chromatin, but rather has a nucleoid structure (containing ssDNA regions) that is dynamically regulated and transcribed. Since CHEX-seq priming is limited by the interval of single-stranded regions and the mitochondrial genome is only ~16 kb, we do not expect many priming events per genic region per mitochondrion. However, there are usually hundreds of mitochondria per cell and the amount of mitochondrial gene transcription varies between cells, hence CHEX-seq should be able to identify mitochondrial transcribed genes. Indeed, we detected ~7.5 mt-genes with ssDNA detected by CHEX-seq in untreated single K562, and the priming count per gene ranged from 1 to 651 (average 73) (Supplementary Fig. 11). For bulk K562 samples, the numbers got even higher: on average 24 single-stranded mt-genes and 8,570 priming counts per gene; the latter is not surprising because there are as many as 600 cells in a bulk K562 sample.

To assess the single-stranded openness along the mt-genome, we calculated CHEX-seq mitochondrial priming density in mouse neuronal cell samples. We observed a nonrandom distribution of priming sites along the circular genome with selected genic regions showing much higher level (≥2 SD) of ssDNA, in cells in the in situ tissue section (Fig. 6a): mt-Co1, mt-Nd2, mt-Nd4l, mt-Nd4, mt-Nd5, mt-Nd6. More interestingly, most of these were shared across the neuronal samples. The astrocytes showed fewer CHEX-seq mitochondrial reads, but several areas were correlated with similar ssDNA regions in the neurons, including mt-Nd2, mt-Co1, mt-Nd4, mt-Nd5. This is consistent with prior evidence showing that neurons use more OXPHOS pathways (reflected in more mitochondrial transcription) to generate ATP than do astrocytes[62,63]. It is intriguing to note that Map2 immuno-identified cells in the mouse neuron sections and Parvalbumin-positive cells in the mouse interneuron tissue section samples showed more mitochondrial ssDNA as compared with primary cultures. Perhaps this is attributable to in situ tissue localized cells being more responsive to their local tissue environment than dispersed cells in primary culture but is also consistent with data showing the PV interneurons have higher metabolic needs[64,65].

Mammalian mitochondria contain a triplex structure called "D-loop", which is formed by the displacement of the heavy strand due to the newly synthesized heavy strand annealing to its template[66–68]. We asked whether the D-loop has more single-stranded openness for CHEX-seq priming. We calculated the per-base priming rate (i.e.,

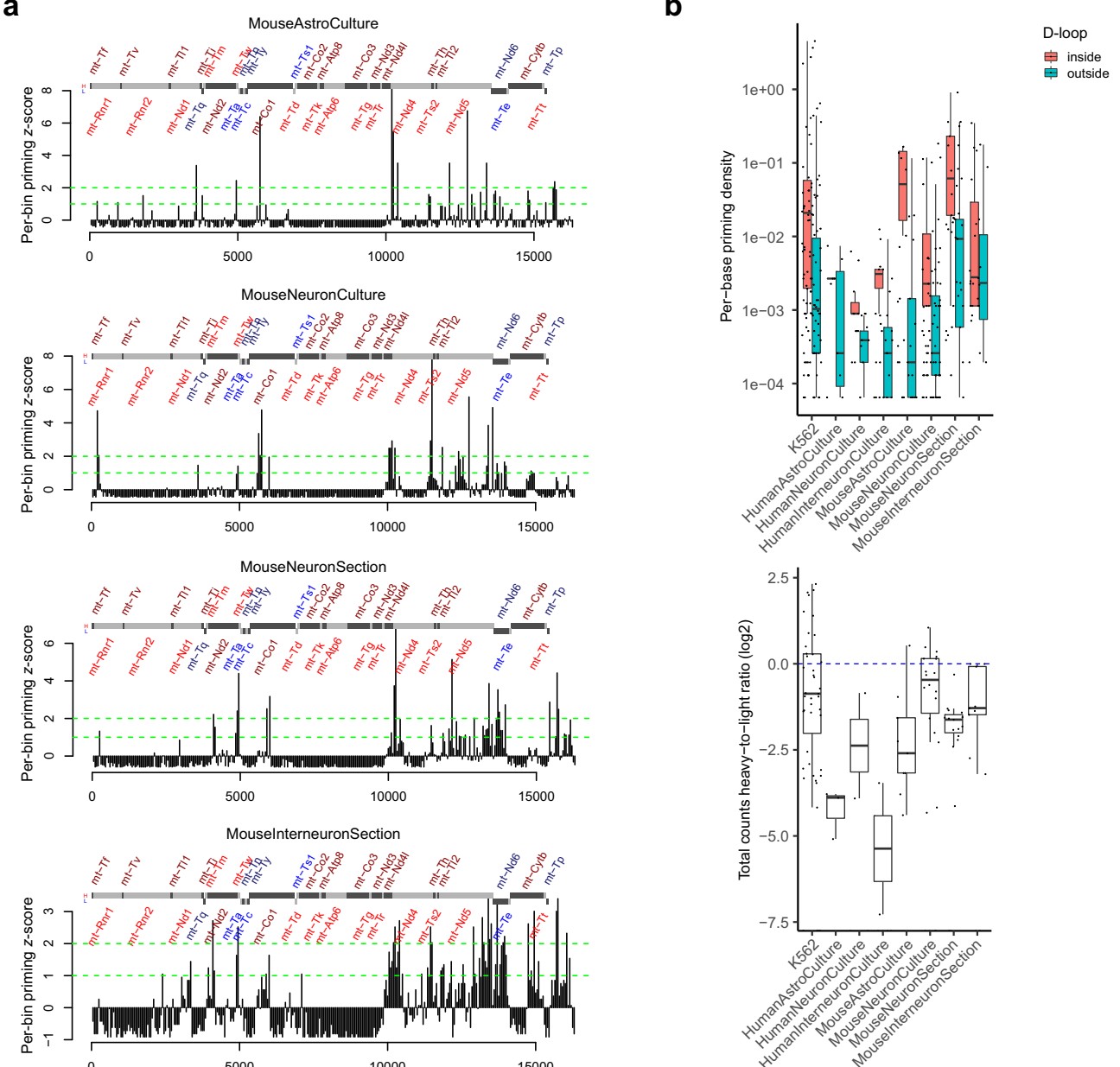

**Fig. 6 | CHEX-seq unveiling the mitochondrial ssDNA in mouse brain.**
**a** Mitochondrial single-stranded hotspots found in mouse astrocytes and neurons in primary cultures, and mouse neuron and interneuron in brain sections. X-axis: coordinate (bp) of the mitochondrial genome; y-axis: per-bin priming counts (z-scored, bin size 50 bp). Green dashed lines highlight where z-score equals to 1 or 2. The red font indicates the heavy-strand genes (upper gray boxes), and the blue font indicates the light-strand genes (lower gray boxes); the dark-light alternating font color corresponds to the shade of the gray boxes, to make adjacent genes more discernible; **b** Top: CHEX-seq priming density (y-axis) inside (red) or outside (green) the D-loop; bottom: CHEX-seq priming bias as measured as the ratio of the heavy-strand counts to the light-strand counts (y-axis); $n = 81, 7, 12, 20, 38, 114, 28, 21$ biologically independent samples for each group (left to right), respectively. The bounds of the box represent the 1st and the 3rd quartile; the thick bar represents the median; the vertical line extends 1.5 times the IQR; the dots represent all data points including maxima and minima.

counts normalized to the length) inside and outside the D-loop for human and mouse samples. Indeed, we found consistently higher priming rate inside the D-loop than outside it for both species and all cell types (Fig. 6b, top).

As noted, CHEX-seq probes preferentially bind to the strand opposite the template strand that gives rise to the natural, sense-strand mRNAs, resulting in a bias towards the antisense-strand reads (Fig. 3d). Since most of the mt-genes ($n = 28$) are located on the heavy strand (i.e., the light strand being the template strand), we hypothesized that the heavy strand should be less occupied by the polymerase, which should allow for more CHEX-seq probes binding, thus leading to

an excess of CHEX-seq reads mapped to the light strand. To test this model, we computed the heavy-to-light strand ratio of CHEX-seq counts. Like K562, all human and mouse data showed systematic bias towards the light strand (Fig. 6b, bottom; Supplementary Fig. 12), which matched our prediction.

### Genomic ssDNA can act catalytically to metalate porphyrin in vitro
Apart from passively serving as a genomic template, we sought to test whether endogenous ssDNA could conduct active functions. There is precedent for endogenous single-stranded nucleic acids having

catalytic activity as some endogenous single-stranded RNAs can act enzymatically and are so-called "ribozymes"[69,70]. Ribozymes are transcribed from genes in the genome but can be engineered in vitro[71] via the systematic evolution of ligands by exponential enrichment (SELEX)[72]. Likewise, in vitro synthesized ssDNAs ("DNAzymes") are able to catalyze various reactions including cutting and modifying RNA molecules[73–75] as well as modification of small molecules such as porphyrin metalation[76]. No endogenous DNAzymes have been described as gDNA has generally been thought to exist predominantly in double-stranded B-form that precludes the DNA from acting catalytically. In assessing potential functionality of CHEX-identified ssDNA, we screened these areas for sequence similarity with published DNAzyme sequences. In human and mouse genome respectively, we identified ~300 and ~900 single-stranded loci with high homology to previously engineered synthetic DNAzymes, including those that had been synthesized to metalate porphyrin.

Porphyrins are present in all mammalian cells[77]. They are critical metabolic precursors to heme and are involved in other biological processes including neuroprotection[78–81]. Some metalated porphyrins they are known to bind to DNA[70,71,73]. The binding to DNA can occur through intercalation of the porphyrin macrocycle between the bases or external to the nucleotide rings, dependent upon the type of porphyrin and metal ion. Porphyrins have also been shown to interact with DNA often through intercalation in G4-DNA quadruplexes[82] including direct interaction with Bcl-2 promoter G-Quadruplex[83]. Further, metalloporphyrins appear to bind to ssDNA[84].

Of the 2200 ssDNA regions predicted to contain gDNAzyme (genomic DNAzyme) sequences, a subset of those that correspond to DNAzymes that metalate porphyrin were selected to screen for enzymatic activity. Nine of the 95 identified CHEX-seq predicted porphyrin DNA metalation gDNAzyme ssDNA regions were functionally tested in vitro, with 6 exhibiting catalytic activity and three showing no activity. Three of the genomic regions that showed activity include two from human RPL7AP61 (chr13:97852922–97852939) and TATDN2P1 (chrX:44290573–44290591) and one from mouse Bmpr1a (chr14:34429016–34429034) (Fig. 7).

As synthetic DNAzymes range in size between 20 and 50 nucleotides, we synthesized 49nt sequences from the CHEX-identified ssDNA regions. Previously, metalation of porphyrin with Zinc ($Zn^{2+}$) has been observed with molecularly evolved synthetic DNA sequences and in some cases it has been shown that it can be enhanced by adding lead ($Pb^{2+}$) as a cofactor[85]. In our studies, we incubated genomic DNAzyme (gDNAzyme) candidates with mesoporphyrin IX (mPIX), $Zn^{2+}$ and $Pb^{2+}$ and observed a time-dependent increase in the insertion of $Zn^{2+}$ into mPIX. The kinetic curves plateaued at 8–12 h (Fig. 7a) and revealed statistically significant increases in the rate of metalation (Fig. 7b). Reactions without the $Pb^{2+}$ cofactor showed little activity (Fig. 7a). As a negative control, isolated mouse liver gDNA was also screened for innate catalytic activity. Both native and denatured gDNA showed a slow but steady non-saturated increase in porphyrin metalation, whose rate was higher than that of reactions without $Pb^{2+}$ cofactor (Fig. 7a, c). The fraction of the total gDNA sequence that contains predicted porphyrin metalation gDNAzyme sequences is small, suggesting that this limits the attainable molar concentration of a gDNAzyme in isolated gDNA for testing in the in vitro assay.

As the endogenous ssDNA regions have structures that are constrained and stabilized by inter- and intra-strand base pairing as well ssDNA binding proteins, we tested the effect of sequence constraints upon porphyrin metalation activity. To assess this, a 90nt gDNAzyme oligonucleotide ("long") was synthesized to which a complementary clamping oligonucleotide, which anneals to internal regions of the long oligonucleotide, forcing a short (45nt) single-stranded loop containing the gDNAzyme sequence to form ("short loop"). A second clamping oligonucleotide that anneals to the ends of the long oligonucleotide was used to force a long single-stranded 66nt loop to form

("long loop"). The short (49nt) single-stranded oligonucleotide ("short") that had showed porphyrin metalation activity was compared to other TATDN2P1 DNA structural variants. The long construct showed the least amount of metalation activity, whereas the short loop showed increased activity and the long loop showed activity that is comparable to the original short sequence (Fig. 7d).

To determine the endogenous genomic localization of the ssDNA that contain putative gDNAzymes, we performed a sensitive two-oligonucleotide Forster resonance energy transfer (FRET) fluorescence in situ hybridization (FRET-FISH) procedure (Fig. 7e). Given the short size of the gDNAzyme to be detected, this FISH procedure is different from that used to assess longer stretches of CHEX-identified ssDNA (Supplementary Fig. 7). For FRET to occur the donor chromophore on the 3′-end of one oligonucleotide must be close to the acceptor on the 5′-end of an adjacent oligonucleotide with the oligonucleotides separated by 5 bases. Detection of the FRET signal was aided by use of a high-resolution analysis procedure that quantitates user-specified subfractions of the excitation and emission spectra for the FRET pairs. This approach reduces artifacts caused by autofluorescence and other background signals. The in situ DNA localizations of the porphyrin metalating gDNAzyme in mouse Bmpr1a was assessed. The FRET-FISH probe-annealed region (red dots) is shown in three cells localized to the periphery of the nucleus in areas occupied mostly by heterochromatin[86] (Fig. 7e). This is consistent with the fact that many of the CHEX-seq reads originated from the repetitive genome. As expected, not all cells showed signal likely due to dynamics of double-stranded/single-stranded DNA transitions (Fig. 4a).

## Discussion

Light-assisted spatially activated CHEX-seq queries single-stranded open-chromatin DNA regions in single cells and thus is distinct from and complementary to other chromatin analysis procedures, such as ATAC-seq that queries double-stranded DNA, and DNase I analysis that cuts both single and double-stranded DNA. As an interrogator of ssDNA in single cells, CHEX-seq assesses not only nuclear DNA but also the single-stranded open-chromatin status of the mitochondrial genome. As a measure of chromatin openness and surrogate for gene transcriptional activity, the ability to assess the transcriptional potential of fixed cells provides a window into the plasticity that underlies a cell's ability to respond to local cues. This is particularly important in understanding the plasticity of neuronal systems where neurons must respond to environmental influences generated through the activity of synaptically interconnected distal neurons as well as local cellular interactions. As a component of cellular phenotype, coupling chromatin status with immunocytochemical categorization will help link a cell's plasticity with its function. Light activation of chromatin state analysis enables the connection to function to be more concrete using fluorescent biomarkers of physiological function that can be imaged at the same time as when chromatin analysis is initiated.

The strandedness of annealing of CHEX-seq primers to ssDNA, apparent in the upstream of transcribed genes, suggests a scaffolding of proteins on one strand of the DNA (blocking CHEX-seq probe binding). This finding is intriguing given the dynamic back and forth between genes' double- and single-stranded status upon chemical perturbation, which emphasizes the role of chromatin 3D structure in transcriptional regulation. The association between ssDNA and gene expression, DNA replication and maintenance indicates multiple sources of single-stranded chromatin and opens the door to questions concerning how different single-stranded mechanisms coordinate in a living cell to produce and maintain the ssDNA landscape in homeostasis. It is also tempting to hypothesize that the balance between the single and double-strand states of chromatin and the rate of interconversion are important to the degree of plasticity or vulnerability that any cell can exhibit. Knowledge of the live cell structural status of

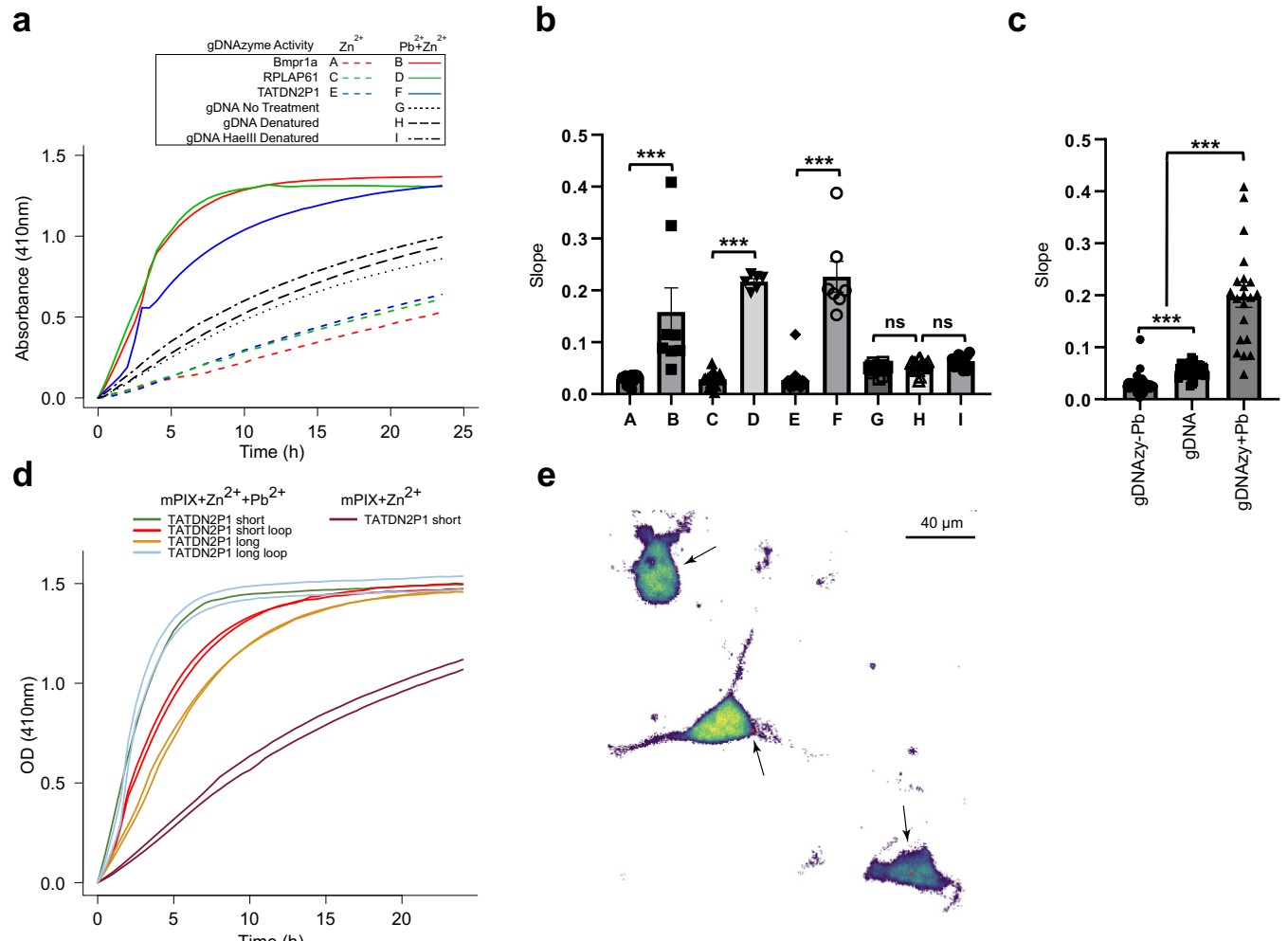

**Fig. 7 | Genomic ssDNA regions can catalyze porphyrin metalation in vitro.**
**a** Insertion of $Zn^{2+}$ into mesoporphyrin IX (mPIX) catalyzed by candidate ssDNA regions of human or mouse gDNA. The plots represent the time course of the absorbance at 410 nm corresponding to the characteristic peak (Soret band) of Zn-mPIX (normalized by the porphyrin concentration). Spectra were captured every 30 min. Solid line: $Zn^{2+}+Pb^{2+}$, dashed line: $Zn^{2+}$ only. Red: Bmpr1a (A,B), green: RPL7AP61 (C,D), blue: TATDN2P1 (E,F); native gDNA (G), denatured gDNA (H), denatured HaeIII cut gDNA (I); **b** Slopes of catalytic activity curves. Data are presented as mean ± SEM. Wilcoxon rank-sum test (two-sided). *ns*, not significant. ***, $p < 0.001$, 3 independent experiments; **c** Comparison of catalytic activity slopes for gDNAzyme w/ and w/o $Pb^{2+}$ cofactor, and mouse genomic DNA. Data are presented

as mean ± SEM. Wilcoxon rank-sum test (two-sided). ***, $p < 0.001$, $n = 3$ independent experiments; **d** Catalytic activity of TATDN2P1 DNAzyme in various conformations. Green: single-stranded short sequence, Orange: single-stranded long sequence, Red: long sequence constrained to form a short loop, Blue: long sequence constrained to form a long loop; Dark red: single-stranded short sequence but w/o $Pb^{2+}$; **e** Mouse cortical neurons in primary cell culture interrogated with FRET-FISH. Black arrows point to the single-stranded DNA areas that contain sequences capable of acting catalytically from the Bmpr1a locus identified by FRET-FISH signal which are visualized with pseudo-colored red dots that are on the periphery of neuronal nucleus. Calibration bar = 20 µm.

gDNA may be useful in designing and targeting therapeutic oligonucleotides and miRNAs that function by annealing to genes to block their expression. Further, as CRISPR targeting of genes for therapeutic DNA editing requires the interaction of the CRISPR complex guide RNAs with ssDNA, target selection may be optimized if genic ssDNA regions are identified or modulated.

The surprising discovery that regions of the endogenous genome have sequence similarity with manufactured DNAzymes and that these sequences exhibit in vitro catalytic activity suggests a novel means for cells to modulate selected biological processes. It is important to note that the in vivo demonstration of DNA catalytic activity remains to be shown. In particular, for the example of gDNAzyme catalyzed porphyrin metalation, the requirement of $Pb^{2+}$ as a cofactor for catalysis of porphyrin metalation suggests that in vivo porphyrin metalation through this mechanism is unlikely as high concentrations of $Pb^{2+}$ are cytotoxic. Normally, one would perform a combination of knockout, mutational and addback experiments to show in vivo functionality but

with the large number of regions showing sequence and structure-constrained porphyrin metalation activity, it is difficult to utilize these standard approaches to elaborate on the in vivo functionality. However, these in vitro functional data show that a subset of CHEX-identified ssDNA loci have the capacity to act catalytically. Any such in vivo activity would require the ssDNA to be in a structure where the active site of the catalytic ssDNA sequence is free to take on a conformation that facilitates its enzymatic activity. The short single-strand porphyrin gDNAzyme shown in Fig. 7d seemingly assumes such a catalytic structure while the long single-stranded oligonucleotide likely exists as a disordered structure that hinders the sequence from acting enzymatically. This hypothesis is supported as when the long oligonucleotide sequence is constrained to produce a ssDNA loop (Fig. 7d) the long ssDNA can act catalytically likely due to assuming a favorable conformation. The longer loop duplex is presumed to have more degrees of freedom to conform to the porphyrin substrate and hence exhibits greater enzymatic activity than the short, more

constrained loop duplex. As the ssDNA landscape dynamically varies between single cells (Figs. 4a and 7e), it is difficult to predict what regions of gDNA may be available to act catalytically at any time. We note that in addition to porphyrin metalation several other synthetic DNAzymes with varying catalytic functions (including RNA cleavage and DNA ligation) have sequence similarity to CHEX-identified single-stranded regions in genomic as well as mitochondrial DNA. It is intriguing to speculate that genomic and mitochondrial gDNAzymes catalyze enzymatic reactions provide locally required compounds. Future studies will determine whether these predicted sequences exhibit in vitro and/or in vivo catalytic activity. Indeed, such single ssDNA regions exist in bacteria and viruses and portend the possibility that they act as gDNAzymes and function catalytically to modulate their biologies.

CHEX-seq also provides evidence for genomic DNA regions that exhibit single-strandedness but are not transcribed, potentially including areas of DNA repair and sites of replication in dividing cells[87,88]. The potential for CHEX-seq to query these other genomic DNA sites awaits future studies and will be enhanced by technical improvements to the CHEX-seq protocol. For example, as the CHEX-seq methodology is refined, it should be possible to multiplex differentially barcoded CHEX-seq probes to transform the CHEX-seq technique into a moderate/high throughput methodology. This would provide spatially defined chromatin landscape information for 10,000's of cells from the same brain tissue section. Further, with selected enzymes and appropriate experimental conditions, it will also be possible to multiplex gDNA chromatin analysis with the transcriptome in the same cell. Finally, the ability of CHEX-seq to work with chemically fixed tissue presents the possibility of analyzing the open-chromatin landscape of individual cells or ensembles of cells in archival human post-mortem fixed tissues from various control and disease states. This capability suggests the intriguing possibility that preserved ancient specimens containing DNA could retain some of their single-stranded chromatin structure (due to preservation), thereby providing the potential for CHEX-seq to define the open-chromatin status of these samples and provide insight into the transcriptome from these ancient cells from which RNA was lost long ago.

Although the present data show the utility of CHEX-seq analysis in chromatin structure research, the current protocol can be optimized in several ways. First, in choosing to prioritize sensitivity of ssDNA region detection the background signal was higher than we would like. We detected a median of 2,640 ssDNA sites in single K562 cells, whilst single-cell ATAC-seq reported a median of 7,125 peaks in K562 (12). The difference may reflect biological differences in the ratio of dsDNA to ssDNA regions in the genome, but may also result in part from a lower CHEX probe photo-activation yield due to the use of the 405 nm uncaging laser line. The optimal 365 nm wavelength was not available for these studies but will produce a 10X increase in uncaging efficiency resulting in more activated CHEX-seq probe, which may increase the number of detected ssDNA regions. CHEX-seq is ideal for analysis of single-strand chromatin in multimodal analyses and would benefit from decreasing the time it takes for the CHEX-seq in situ annealing and in situ transcription. This will be explored with optimized hybridization and reaction conditions that maintain the ability to multimodal analyze other chemicals in the same cell queried by CHEX-seq including proteins and metabolites. Further, while CHEX-seq works on immunostained cells in fixed tissue sections, to analyze the ssDNA chromatin landscape of thousands of in situ localized cells the approach needs to be transitioned into a high-throughput process. As CHEX-seq is a light-activated process, to make it high throughput for single cells will require automated dynamic photomasking which will best be accomplished by developing AI algorithms to control this "on-scope" process. Despite these limitations, CHEX-seq revealed the genome-wide landscape of in situ single-stranded non-B-form DNA, and further established the association of ssDNA in transcribed genes (especially their intronic regions) with gene transcription. It also highlighted the association of ssDNA in distal intergenic areas with genome replication, repair and maintenance. Our data suggests an important regulatory role for the dynamic interconversion and ratio of single- to double-stranded DNA in chromatin dynamics. As such, the ability to perform single-cell, spatial open-chromatin, non-B-form DNA analysis on immuno-identified single neuronal cells promises to provide high-resolution spatial genomic landscape analysis of dynamic circuit function and disease-associated dysfunction in functionally relevant cells.

## Methods

### Human brain tissue
Human brain tissue was collected at the Hospital of the University of Pennsylvania (IRB#816223) using standard operating procedures for enrollment and consent of patients. Briefly, an en bloc sample of brain (typically 5x5x5 mm) was obtained from cortex that was resected as part of neurosurgical procedures for the treatment of epilepsy or brain tumors. This tissue was immediately transferred to a container with ice-cold oxygenated artificial CSF (KCl 3 mM, NaH$_2$PO$_4$ 2.5 mM, NaHCO$_3$ 26 mM, glucose 10 mM, MgCl$_2$-6H$_2$O 1 mM, CaCl$_2$-2H$_2$O 2 mM, sucrose 202 mM, with 5% CO$_2$ and 95% O$_2$ gas mixture) for transfer to the laboratory. Tissues arrived in the laboratory ~10 min post excision. The brain tissues were then processed for cell culturing and fixation (see below). The use of Human Subjects was approved by the University of Pennsylvania IRB. All subjects provided written informed consent for the use of tissue samples. The research was conducted according to the principles of the Declaration of Helsinki.

### Cell culturing/preparation and fixation
K562 cells were obtained from ATCC and cultured in RPMI 1640 medium (Invitrogen) with 10% FBS and penicillin-streptomycin in a T75 flask at 37 °C in 5% CO2 for 2 - 3 days. The cultured cells were transferred to a 50 ml tube and 16% paraformaldehyde (final 1%) was added for 10 min at room temperature to fix the cells. After fixation, 1 M glycine (final 200 mM) with RPMI 1640 medium was used to quench for 10 min followed by centrifugation at 300 x g for 5 min. The supernatant was discarded, and 3 mL of PBS were added to the pellet and then mixed by gently pipetting up and down 10–15 times using a fire-polished glass-pipette, to prevent cell clumping, and centrifuged at 300x g for 5 min. The 100 µl cell pellet was attached to 18 mm gridded coverslips by incubating them for 2 h at room temperature. The samples were treated with PBS (w/o Ca$^{2+}$, Mg$^{2+}$) containing 0.01% Triton X-100 for 10 min and then washed with PBS (w/o Ca$^{2+}$, Mg$^{2+}$) 3 times for 3 min. To prepare human neuronal cell cultures, adult human brain tissue was placed in the papain (20 U, Worthington Biochemical) solution to dissociate at 37 °C for 30 to 40 min and followed by ovomucoid (a papain inhibitor, 10 mg/ml, Worthington Biochemical) to stop the enzymatic dissociation[89,90]. The tissue was triturated with a fire-polished glass Pasteur pipette. The cloudy cell suspension was carefully transferred to a new tube and centrifuged at 300 x g for 5 min at room temperature. The cells were counted in an Autocounter (Invitrogen). Cells were plated on poly-L-lysine-coated (0.1 mg/ml, Sigma-Aldrich) 12 mm coverslips at a density of 3 × 10$^4$ cells/coverslip. Cultures were incubated at 37 °C, 95% humidity, and 5% CO$_2$ in neuronal basal medium (Neurobasal A, Gibco), serum-free supplement (B-27, Gibco) and 1% penicillin/streptomycin (Thermo-Fisher Scientific). Dispersed mouse embryonic neuron/astrocyte cultures were prepared following published protocols[90]. As cells are generated and mixed from embryos of both sexes, sex was not considered in the use of these cultures. Approval for the harvesting of mouse neuronal cells was provided by the University of Pennsylvania IACUC oversight committee (Approved Protocol 801873). Mice were housed at University of Pennsylvania animal facility in the John Morgan Building under a normal day-night cycle. Dispersed mouse neuronal cells that were

cultured for 2 weeks were fixed using 4% paraformaldehyde for 10 min at room temperature. We then performed three washes with 1x PBS. The cells were permeabilized with 0.1% Triton-X100 for 10 min at room temperature followed by another three washes with 1x PBS. For select experiments, K562 cells were treated with 16 mM TPA (12-O-tetra-decanoylphorbol-13-acetate) for 15 min, 1, 2, and 24 h.

## Mouse brain tissue sections

A three-month-old male mouse was anaesthetized with halothane, euthanized by thoracotomy, then subjected to cardiac perfusion with 5 ml PBS followed by 20 ml PBS/4% paraformaldehyde. The brain was removed and post fixed at 4 °C for 16 h, then rinsed in PBS and sectioned in the coronal plane at 100 μm on a vibratome (Leica VT-1000s). Sections including the hippocampus were then subjected to immunofluorescence labeling with chicken anti-MAP2 antibody (1:1000; Ab 5392; Abcam) followed by Alexa 488 conjugated goat anti-chicken secondary antibody (1:400; ab150169; Abcam).

## CHEX-seq probe synthesis

HPLC-purified probe oligo and its complimentary oligo were purchased from Integrated DNA Technologies (IDT). A template-dependent DNA polymerase incorporation assay was employed to extend Cy5-dye-labeled Lightning Terminator™ (Agilent, Inc.) to the 3' end of probe oligo: (1) 5 μM of probe oligo, 25 μM complimentary oligo, 50 μM of Cy5-labeled Lightning Terminator™, 4 mM MgSO4, and 0.1U/μL of Therminator (New England Biolabs) were mixed in 1x ThermoPol buffer, (2) the mix was heated to 80 ˚C for 45 s and (3) then incubated for 5 min at each of 60 ˚C, 55 ˚C, 50 ˚C, 45 ˚C, 40 ˚C, 35 ˚C, 30 ˚C and 25 ˚C. The incorporation product was purified on the 1260 Infinity reverse phase HPLC (Agilent Technologies) using the Xterra MS C18 Prep column (Waters). The purified product solution was concentrated to approximately 250 μL using the Vacufuge (Eppendorf) followed by denaturation into single-stranded oligo with equal volume of 0.2 M NaOH. HPLC purification and concentration were repeated using the same conditions for collection of the Lighting Terminator-labeled single-stranded probe. The final product was dissolved into 1xPBS, and the concentration was determined by measuring Cy5 absorbance at 647 nm (Supplementary Fig. 1).

## CHEX-seq probe annealing, imaging and photoactivation

After fixation and permeabilization, the cells and brain sections were incubated with CHEX-seq probe (170 nM) in TES buffer (10 mM Tris, 1 mM EDTA, 150 mM NaCl) for 1 h at room temperature. The samples were then washed with 1x PBS (w/o Ca²⁺, Mg²⁺) 3 times for 3 min. After CHEX-seq probe annealing and washing, the samples were transferred to the imaging chamber with 1x PBS (w/o Ca²⁺, Mg²⁺). All images and photoactivations were performed using a Carl Zeiss 710 Meta confocal microscope (20x water-immersion objectives, NA 1.0). CHEX-seq probe annealing was confirmed by exciting at 633 nm and emission was detected at 640–747 nm. The photoactivation was performed using the 405 nm (UV) laser at 60% power and 6.30 μs per pixel.

## First-strand DNA synthesis in situ and single cell harvesting

After photoactivation in each individual cell's nucleus, a master mix containing DNA Pol I and 1st strand DNA synthesis buffer was added to the cells and incubated for 1 h at room temperature. Subsequently, the single cells containing synthesized cDNA were harvested using a glass micropipette under using a Zeiss 710 confocal microscope (Carl Zeiss) for visualization.

## Linear amplification of nucleosome-free areas of chromatin

(A) 1st strand DNA synthesis and poly G tailing at 3' end: After harvesting single cells, the in situ synthesized cDNA was removed by adding fresh prepared 0.1 N NaOH and incubating the sample for 5 min at RT followed by neutralization with 1 M Tris (pH 7.5). After ethanol

precipitation, the 1st strand DNA was resuspended in nuclease-free water. Subsequently, poly(G) was added to the 3' end using terminal deoxynucleotidyl transferase (TdT) (Invitrogen). (B) 2nd strand DNA synthesis and round 1 linear RNA amplification: 2nd strand DNA was synthesized using DNA Pol I for 2 h at 16 °C after priming with custom App-RC-polyC primer (Supplementary Fig. 1). RNA was amplified using linear in vitro transcription from T7 RNA polymerase promoter incorporated into the double-stranded DNA with Ambion MEGAscript T7 In Vitro Transcription (IVT) Kit. (C) Round 2 1st and 2nd strand DNA synthesis and PCR amplification: After cleanup IVT reaction, 1st strand DNA was reverse transcribed from aRNA using Superscript III using a custom App-RC primer and 2nd strand DNA was synthesized using DNA Pol I with a custom XXbpPBCYY primer (XX = 16–18, YY = 1, 15–24; for the complete list see Supplementary Table 1). Subsequently, the double-stranded blunt-ended DNA was amplified using custom primers XXbpPBCYY / App-RC (Supplementary Table 1) following PCR condition: 98 °C for 30 s; thermocycling at 98 °C for 10 s, 50 °C for 30 s, 72 °C for 30 s for 27 cycles; extension at 72 °C for 2 min and was then used for library construction. To distinguish CHEX-seq priming from endogenous PCR priming, the PCR primers were designed with their 3' ends clipped up to 4 bp. Samples for the control experiments were processed with the same procedure except no CHEX-seq probe was applied, and 2nd round 2nd strand DNA PCR amplification was performed with custom primers 18bpPBC14 / App-RC (Supplementary Fig. 2).

## Sequencing library preparation

Illumina TruSeq Nano DNA Library Preparation Kit was used with modifications. The entire second round PCR amplified double-stranded DNA was used as input. After converting DNA fragment into blunt ends with End Repair Mix, base A was added; sequence adapters were ligated. DNA inserts were amplified with PCR. High-throughput sequencing was done on Illumina NextSeq 500. (For a complete list of sequenced libraries see Supplementary Data 1).

## Data preprocessing

Raw FASTQ mates were concatenated by the read pair, then multiple copies of the same concatenated sequence were reduced to a unique one. Any identical read pairs from two different samples were defined as the shared reads and removed. Substrings from the primer 2p and pC were searched and trimmed (if found) sequentially (first 2p, then pC) (Supplementary Fig. 2, Supplementary Table 1). After trimming, reads shorter than 10 bp were discarded. Subsequently, poly Ns were trimmed (if found) from the ends. The resultant reads shorter than 10 bp were discarded. Read pairs were then aligned by STAR[91] to the GENCODE GRCh38 genome[92,93] (for human samples) or the UCSC mm10 genome[28,29] (for mouse samples), with splicing junction disabled. The joint set of mapping criteria are (1) minimal mapping length 10; (2) minimal score normalized to read length 0.4; (3) minimal matched bases normalized to read length 0.4; (4) maximal mismatched bases normalized to mapped length (i.e., per-base mismatch rate) 0.1. The alignments were further filtered for the primary location and non-PCR duplicates called by Picard (https://github.com/broadinstitute/picard). Reads with a non-overlapping mapped length below 20 bp or mismatch rate greater than 0.1 were excluded.

To eliminate possible contamination, each human or mouse sample was aligned to alternative genomes: for human samples, mouse, bacteria (top 20 species) and mycoplasma; for mouse samples, human, bacteria (top 20 species) and mycoplasma. For each sample of a given species, the reads that aligned better to an alternative genome than the target genome were defined as contaminants and removed. To remove misalignments, we synthesized 20 bp single-end reads and mapped them to human or mouse reference using STAR and the same parameter setting. We called genomic regions where reads could not be faithfully mapped (i.e., to a wrong chromosome or to the same

chromosome but >10 bp off) and then filtered them out from the CHEX-seq reads. In addition, human and mouse blacklist regions were obtained from ENCODE[94], and CHEX-seq reads whose barcode overlap with the blacklist regions were excluded.

## CHEX-seq ssDNA calls and priming counts

Based on the presence and format of 5' end T7 barcode and 3' end pC primers, the alignments were classified into four categories: A, B, C, and D, among which A to C were divided into subclass 1 or 2, depending on the presence of read-through or not (Supplementary Fig. 3, Supplementary Data 2). For the sake of a balanced sensitivity and specificity, class A (both proper 2p and pC primer) and B reads (proper 2p/barcode primer only) were defined as good quality and merged as the AB reads. Then barcode locations were tracked in 20 bp bins across the genome. For each sample with $k$ cells, the number of reads whose barcodes hit the same bin were reduced to a maximum $2 \times k$, given the diploidy of the samples in this study. We prioritized the barcode/primer quality (A1 > A2 > B1 > B2 > C1 > C2) and the mapping quality (STAR alignment score, mapped length) when deciding the duplicate priming reads. We examined the data quality at each barcode/primer criterion threshold (A only, A and B merged, A, B and C merged) (Supplementary Data 3) and chose the A, B and C merged processing for the sake of maximal sensitivity given the sparsity of CHEX-seq priming. To evaluate the consequence of the inclusion of the C reads, we performed comprehensive analyses and showed that the impact of relaxed barcode/primer quality selection is negligible (for details see Methods: Background Estimation and Processing). To count gene-level priming frequency, we defined various genic regions and sub-genic regions (Supplementary Fig. 10a), based on Ensembl gene models from R packages EnsDb.Hsapiens.v86 (human) and EnsDb.Mmusculus.v79 (mouse)[95]. For CHEX-seq priming site annotation, we used R package GenomicRanges[96] and ChIPseeker[97]. The GO enrichment analysis was done with R package clusterProfiler[98,99].

## Background estimation and processing

To estimate the false positives we performed three types of control experiments in K562: (1) with barcoded probes but no laser activation (Probe(+) Laser(-)), (2) neither barcoded probes nor laser activation (Probe(-) Laser(-)), (3) with barcoded probes and laser activation but samples were mung bean digested (MungBean). Using the Probe(-) Laser(-) controls, we estimated the background in non-control samples as 14.4% and 27.6%, for single-cell or bulk K562 (Supplementary Fig. 4a, left). MungBean controls exhibited a higher background, due to an incomplete digestion of the ssDNA. Only Probe(+) Laser(-) samples showed surprisingly more reads than non-controls. Further dissection of the read class revealed that the signals in control samples arose almost all (>90%) from C reads, i.e. the reads without the 5' barcoded primer (2p) but with the 3'primer (pC), whilst the fraction was much smaller for non-control samples (Supplementary Fig. 4a, right). We also inspected the sequence complexity of aligned reads in Probe(+) Laser(-) samples and observed a much lower complexity (-0.5) as compared with the non-control samples (-0.68). As most of the low-complexity controls carried the probe 517 s (Supplementary Fig. 4b), we suspected that their unusually high background was something specific to this probe.

We performed a detailed evaluation of the impact of the background. First, we examined the TSS coverage and found that only non-control K562, including or excluding the background, showed TSS peaks as an aggregate, although the pattern became weakened for individual cells (Supplementary Fig. 4c–e). Consistent with the disappearance of TSS peaks, we found that compared with the non-controls, Probe(+/-) Laser(-) exhibited a 33.3% increase (from 60.1% to 80.6%) in distal intergenic reads and a 61.1% decrease (from 12.2% to 4.8%) in promoter reads (Supplementary Fig. 4f). Next, we compared Probe(+/-) Laser(-) and non-control samples in their enrichment in

ENCODE ssDNA related epigenomes. Non-control K562, including or excluding the background, showed significantly higher ($p = 0.083$ or 0.0096, respectively) fold-of-enrichment than Probe(+/-) Laser(-) samples (Supplementary Fig. 4g, left). Lastly, we compared Probe(+/-) Laser(-) and non-control samples in their enrichment in K562 transcriptome (RNA-seq or GRO-seq), and we observed ubiquitously stronger fold-of-enrichment in non-control samples than controls for all genic/sub-genic regions and for both RNA-seq and GRO-seq datasets (Supplementary Fig. 4g, right).

## Porphyrin metalation DNAzyme homologous ssDNA loci

We downloaded catalytic core sequences of the synthetic DNAzymes previously shown to be active in porphyrin metalation from the database DNAmoreDB[100]. We then aligned them to the human and mouse references using BLAST with the criteria (1) 85% mapped length, (2) 0 mismatch and (3) overlapping with at least one CHEX priming site. This led to 326 and 904 porphyrin metalation homologous ssDNA loci being identified in human and mouse, respectively (Supplementary Data 6).

## Mesoporphyrin metalation assay

All standard reagents and solvents were purchased from commercial sources and used as received. Mesoporphyrin IX (mPIX) was purchased from Frontier Scientific. UV-vis absorption spectra were recorded on a Lambda 365 UV-Vis spectrophotometer (Perkin-Elmer). A quartz micro-cuvette (Starna Cells, 0.1 ml) was used for spectroscopic measurements. All aqueous solutions were prepared using deionized water (dH$_2$O). SB buffer consisted of Tris (100 mM), NaOAc (200 mM), KOAc (25 mM), Mg(OAc)$_2$ (10 mM), Triton X (0.5% by weight) and DMSO (5% by volume). pH was adjusted to near-neutral (-7.2) using NaOH (12 N) or HCl (6 N). Stock solutions of mPIX in DMSO (15 mM), Pb(OAc)$_2$ in dH$_2$O (50 mM) and Zn(OAc)$_2$ in dH$_2$O (50 mM) were used to prepare mixtures for kinetic experiments. Kinetics of Zn$^{2+}$ insertion into mPIX was monitored using optical absorption spectroscopy. All experiments were conducted at 23 °C. A solution for spectroscopic measurements was prepared directly in a micro-cuvette. The cuvette was charged with SB buffer (100 μL), and all components were added to it by a micro-pipette using the respective stock solutions. The final concentrations were: mPIX (1.5 μM), Pb(OAc)$_2$ (1 mM), Zn(OAc)$_2$ (1 mM), DNAzyme (1 μM) or gDNAzyme (25 ng/100 μL). After addition of the last component (typically Zn(OAc)$_2$), absorption spectra were recorded every 30 min during 24 h. Zn$^{2+}$ insertion was monitored by measuring the absorbance at the maximum of the Soret band (410 nm) corresponding to Zn-mPIX. To verify that the change in the absorbance was indeed due to the formation of the target complex (Zn-mPIX), the Q-band region was examined in the end of each kinetic run, confirming that the absorption in the Q-band region was consistent with that in the Soret region. Each kinetic measurement was repeated three times to ensure reproducibility.

## In vitro gDNAzyme catalytic kinetics

The activity of the enzyme was calculated by normalizing the time course of the measurement (see Fig. 7a, d(P)/dt) to the maximum rate (Vmax) and calculated the time constant of activity (tau) as half the maximum rate for the entire measurement (see Fig. 7b, c). We also measured the time constants for each segmented measurement (30-minute measurements over 24 h) and observed a statistical significance for the different DNAzymes (see Fig. 7b, c).

## In vivo localization and analysis of gDNAzyme loci using FRET-FISH

The FRET-FISH Probes for DNAzyme were designed based on two FISH probes couples to two fluorescent dyes (ATTO550/ATTO590) with overlapping spectra targeting two proximal DNA sequences that close enough to generate FRET that can be detected (Supplementary

Table 2). The labeled probes are 21–25 bases long and are 5 bases away with the ATTO590 facing toward the ATTO550. The proximity of ATTO550 and ATTO590 generates a FRET signal that shows the spatial location of predicted gDNAzymes inside cells. Primary mouse neurons were cultured at 12 mm coverslip[90]. On the day of hybridization, the cells were rinsed in 1xPBS/0.12 M sucrose (pH7.4) and fixed with fresh prepared 4% formaldehyde (PFA)/1xPBS/0.12 M Sucrose for 30 min at RT. Subsequently, the cells were quenched in 0.1 M glycine for 10 min to remove unreacted PFA and washed in 1xPBS (pH7.4) for 5 min 3 times. The cells were permeabilized in 0.1% Triton X-100 for 10 min RT and washed again in 1xPBS (pH7.4) for 5 min 3 times.

The FISH hybridization mix was prepared by mixing the FRET oligonucleotide probe pairs in hybridization buffer (4x SSC, 0.5 mM EDTA, 10% formamide) at final concentration 100–200 uM, and added to each cell for overnight reaction at RT. The next day, the cells were washed sequentially with 4x SSC for 5 min 2 times, 2xSSC for 5 min 2 times, 0.5xSSC for 5 min 3 times, and distilled water once. The cells mounted on glass slide in Vector mounting media containing DAPI. The cells were screened for FRET detection using a Zeiss 880 confocal microscope. To detect low-level fluorescence signals from FRET with high confidence, the emission signals were separated by spectrum using a grating. Both acceptor and emission signals were spectrally separated in 15 nm resolutions, divided into four channels each, and integrated using a multichannel GaAsP PMT. In contrast to conventional FRET quantification using two channels, four channels for both donor and acceptor emission signals were used to evaluate FRET efficiency. The mean fluorescence signal of each channel in the cell of interest was calculated and compared to the fluorescence signal changes of each pixel. Any increase or decrease in the signal that was more than defined Z standard deviations from the mean value of every pixel fluorescence value in the cell of interest was considered significant. To determine successful FRET in single pixels, the value of each channel was computed and all four channels in the donor or acceptor had to show either a decrease or increase respectively. This process not only increased the detection sensitivity of FRET at the level of single pixels but also significantly decreased the likelihood of false positives due to unavoidable system noise.

### Reporting summary
Further information on research design is available in the Nature Portfolio Reporting Summary linked to this article.

## Data availability
The raw CHEX-seq data (in FASTQ format) and the processed CHEX-seq data generated in this study, including the per-sample priming sites (in BED format) and the gene-by-sample priming count matrices, have been deposited in the GEO database under accession code GSE231719. The mouse brain single-cell RNA-seq data from primary culture and from acute slice have been deposited in the GEO database under accession code GSE231725. K562 and human brain transcriptomes, and various epigenomes from K562 and human/mouse brain curated from public databases and being re-analyzed in this study have been deposited in the GEO database under accession code GSE232215. The dbGAP data for the human CHEX samples is available as phs002120_v1 (https://www.ncbi.nlm.nih.gov/projects/gap/cgi-bin/study.cgi?study_id=phs002120.v1.p1 and https://ftp.ncbi.nlm.nih.gov/dbgap/studies/phs002120/phs002120.v1.p1/). The sample sheet and results of the main analyses are available in Supplementary Data. Source data are provided with this paper.

## Code availability
CHEX-seq NGS pipeline is freely available at https://github.com/kimpenn/chex-seq [101]. Data analysis scripts are public at https://github.com/kimpenn/chex-analysis [102].

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

## Acknowledgements

We thank Eun-Hee Shim for technical help in the early stages of this work and Kevin Miyashiro for culturing the mouse neuronal cells. Jai-Yoon Sul helped with the FRET-FISH analyses as well as performing the gDNA-zyme enzymatic activity calculations. Dr. Jinchun Wang and Mimi Healy, Lasergen synthesized and provided the CHEX primers.

## Author contributions

J.L., J.Li, J.Wa, J.E. conceived of and performed the molecular experiments; S.R.A. and S.V. performed the porphyrin metalation assays. Y.L., K.L. B., S.F., H.K., C.E.N., J.Y.K. conceived of and performed data analysis; A.U., S.B., H.C., J.Wo., M.G. provided human neurosurgical tissue; S.A. provided immunostained brain sections; J.Y.K., J.E. conceived of experimental paradigm; J.Le, Y.L., J.Wa., J. Li., K.L.B., S.F., C.E.N., J.R., S.A., A.U., S.A.R., S.V., S.B., H.I.C. J.Wo., M.S.G., J.K. J.E. wrote and reviewed the manuscript.

## Competing interests

The Authors declare the following competing interests. J.E., J.Y.K., Y.L., S.F., J. Li, are co-inventors on a published patent application (20200216841) covering the CHEX-seq technology applied for by the University of Pennsylvania and Agilent Technologies. J.E., Y.L., J.Y.K., S.R.A., S.V. are co-inventors on a provisional patent application (63/511,984) covering gDNAzymes applied for by the University of Pennsylvania. The remaining authors declare no competing interests. This work was funded in part by NIH U01MH098953 (J.E., J.K.) and RM1HG010023 (J.K., J.E.), and by R01MH110185 (S.A).
