## [Peer Review File · Nature Communications]

CHEX-seq Detects Single-Cell Genomic Single-Stranded DNA With Catalytical PotentialREVIEWER COMMENTS

Reviewer #1 (Remarks to the Author):

Overview: The authors describe a novel sequencing method to identify single stranded DNA (ssDNA), which they call CHEX-seq (Chromatin Exposed -seq). The technology is currently under application for a patent by several of the authors. This research team demonstrated the efficacy of CHEX-seq compared to similar chromatin assays, such as ATAC-seq, in primary cell cultures and its application to spatial profiling in mouse brain sections. The authors also identified some endogenous ssDNA sequences which had catalytic activity for porphyrin metalation, suggesting they potentially discovered a novel molecular function of genomic ssDNA.

There are a number of questions that comes to mind when contemplating the application of this novel technology. For example, is CHEX-seq data analysis capable of differentiating the current activity occurring at ssDNA loci? In other words, is a particular loci ss because it is being actively transcribed or actively replicated? Is it being accessed by a specific DNA repair mechanism? After reading through the manuscript, it appears that the authors fail to adequately address this question. For example, cross-referencing CHEX priming location with respect to transcription start sites with gene expression data from RNA-seq or GRO-seq. They also found that a strand bias towards antisense strand reads was strongest near the transcription start sites and postulate a reasonable explanation. However, they mention in the Discussion that, while there is evidence for CHEX-seq's ability to detect single stranded regions that are not being transcribed (such as sites of DNA repair and replication) additional technical optimizations and improvements in CHEX-seq technology will be required.

Minor Concerns:

1. There are several typos throughout (for example ATAC-see instead of ATAC-seq), so some proofreading is necessary.
2. In general, the methods are thorough and comprehensive. The authors provide GitHub repositories sharing their analysis pipeline and scripts.

Overview of CHEX analysis and experimental design: the technology works by using a degenerate sequence which anneals to ssgDNA. The degenerate sequence has a 5' barcode for single cell multiplexing and a 3' fluorescently tagged deoxynucleotide which can be activated with light (meaning the probe will function as a primer for cDNA synthesis when photoactivated). In this case, they use the 405nm laser on a confocal microscope for photoactivation. They then incubate the cells in the cDNA synthesis buffer and harvest individual cells manually with a glass micropipette.

Benchmarking in K562 cells: There seems to be a lot of priming sites indicated in the controls despite no CHEX-seq probes. This deserves some mention and explanation in the text. Presumably these "background" priming sites would not be barcoded and therefore could be subtracted? It is unclear just how the authors handle this background in the data analysis. In the methods they mention distinguishing CHEX priming from endogenous PCR priming using clipped 3' ends, but I failed to find specific mention of the non-CHEX priming sites in the priming counts section of the methods. This requires clarification in any revision of this otherwise excellent manuscript.

Comparison to other open chromatin assays: The comparison is pretty robust, especially when they extend their analysis to 284 K562 epigenomes across multiple assay types. The use of the FISH assay to validate single strandedness of specific loci provides good supporting validation for CHEX-seq's ssDNA specificity.

The authors spend most of the Discussion describing the potential applications for this technology, which are wide-ranging and clearly significant. Overall, this is will undoubtedly be an impactful

technology that will contribute to multiple discoveries in the biological sciences over the next decade or more. However, it is my sense that the manuscript lacks a genuine analysis of the current limitations of CHEX-seq. Where are the current deficiencies or limitations in the technology that hinder its applicability, and in what ways does it need to be improved and optimized? For example, they mention a surprising number of these background priming sites in controls that didn't even receive CHEX probes? Based on the numbers reported in the K562 cells between negative controls and CHEX-seq processed cells, it looks like the number of non-CHEX priming sites may account for as much as 17% of total priming sites (14,704 in bulk control samples compared to 85,382 in bulk CHEX samples). Looking at the medians they report in single cells, they may account for 28% (748 of 2,640) of priming sites in the single cell experiments. It is important to note that the authors state that this was what the data looked like after QC filtering. It would be beneficial if the research team would further explore this question and try to explain why there are so many non-CHEX priming sites and what are the implications of this for the quality of CHEX-seq data. On the same note, in one section they mentioned that they performed a modified analysis by adjusting the barcode and primer class criterion as well as the alignment criterion in order to reduce spurious reads. Can the authors calculate an estimation for the proportion of total reads that could be spurious? In spite of these questions, the manuscript has every likelihood of being very impactful to the life sciences and deserves publication.

Reviewer #2 (Remarks to the Author):

Lu et al described a new method, termed CHEX-seq, to detect single cell Genomic ssDNA with catalytical capacity. It is an interesting method, and the manuscript is well written and organized. Some of issues need to be addressed before it is published.

1, In the abstract and main text, they claimed they used formalin-fixed cells, actually it is 4% formaldehyde instead of formalin, which is different from formalin. The description has to be accurate. Otherwise, they need to prove: CHEX-seq works in formalin-fixed paraffin embedding clinical samples.

2, In the workflow of CHEX-seq, No RNaseA or the other enzyme was used to digest native RNA in cells, would that be the issue for the potential contamination of CHEX-seq?

3, When replication happens in the genome, ssDNA could be captured. Indeed, Figure 1 B, RepOrig is in the same branch of FAIRE, DNASE, ATAC. Could author compare CHEX-seq data with RepOrig data in genome wide to learn something?

4, Figs. 1 and 7: Author Showed that with a close comparison between the two assays (Fig. 1D and EX5A). CHEX-seq differs from ATAC-seq in having a wider peak with an extended slope of 5' to the TSS. It means that CHEX-seq captures additional information regarding transcriptional initiation events. In vitro, it has been shown that SSDNA is involved in catalyzing porphyrin metalation (Fig. 7). Could you clarify the evidence linking in vivo (Fig. 1) and in vitro (Fig. 7) events that CHEX-seq captures the SSDNA events other than DNA replication and associated events such as DNA methylation maintenance?

5, Fig. 3: As it is mentioned in the K562 cells (Fig. 1C), the highest proportion of priming sites is found in the intergenic regions (> 60.1%), followed by introns (~26.1%), and then by promoter regions (less than 5kb to the TSS) (~12.2%).

So, it means more than 86% of CHEX-seq priming sites are located in intergenic sites and non-coding

parts of the genome other than the coding sequence. Does it mean that introns and intergenic sites of the genome are more likely to have it in the SSDNA state? Does CHEX-seq predict the most frequently occurring enzymatic activity by using these SSDNA originating from intergenic sites and introns?

6, Fig. 7: As you have mentioned, endogenous SSDNA have sequence similarity with manufactured DNazymes, and these sequences exhibit in vitro catalytic activity, but you found only three of them in the genome that have in vitro catalytic activity. What about the rest of the other sequences that failed to show enzymatic activity? Could you elaborate in detail? Could author provide supportive direct evidence for SSDNA catalyzing enzymatic activity in vivo?

Response to Reviewers:

We thank the reviewers for their considered and helpful reviews. We have modified the paper in accord with their suggestions.

General Revisions - We corrected the spelling of one of the authors names (Bullaughay). We have also updated Fig 6 to reflect the per-bin priming z-score, Fig 7 to be more legible and corrected the color keys, and Extended Data Fig 4 to address comments of the reviewers.

Changes to the text are highlighted in yellow.

Response to Reviewer 1's comments:

We appreciate the reviewer's careful review and constructive comments. Here, we address the reviewer's questions point by point.

- *is CHEX-seq data analysis capable of differentiating the current activity occurring at ssDNA loci? In other words, is a particular loci ss because it is being actively transcribed or actively replicated?*

We believe that, active transcription or DNA replication, both contribute to the genesis and maintenance of ssDNA. On one hand, the well-established "transcription bubble" model (Bieberstein et al. 2012, Barnes et al. 2015) indicates the existence of short single-stranded regions in actively transcribed loci. In our experiment data, we did observe a significant overlap between CHEX-seq sites and highly expressed genes, and the overlap got even stronger when compared to GRO-seq detected nascent transcripts. On the other hand, we also observed relatively close distance between CHEX-seq and Replication Origin (RepOrig) data at 5kb resolution (Fig 2B), which is in line with the anticipated single-stranded openness at the replicating DNA. Besides, we found that several genome repair/maintenance proteins, such as the Y-box family and the mini-chromosome maintenance (MCM) family, were specifically enriched in CHEX-seq than ATAC-, DNase- or FAIRE-seq (Fig 2C). Further, our observation of the DNA methyltransferase DNMT1, whose genomic enrichment was significantly higher in CHEX-seq than in other open-chromatin assays (Fig 2C), adds to the evidence that ssDNA is involved in genome replication/maintenance, since DNMT1 is known for methylation inheritance from the template strand. Thus, we established the connection between ssDNA and active transcription or DNA replication/maintenance.

For a particular locus, however, we are not able to distinguish the source of ssDNA because our current protocol of CHEX-seq detects ssDNA from all underlying mechanisms as a whole at the single-cell level.

- *Is it being accessed by a specific DNA repair mechanism? After reading through the manuscript, it appears that the authors fail to adequately address this question.*

As clarified in the previous point, we attributed a significant portion of the observed ssDNA to genome replication/repair/maintenance activities. However, since the evidence largely came from comparing CHEX-seq and ENCODE ChIP-seq data, we are able to conclude an association between ssDNA and DNA repair/maintenance TFs in a general sense, but we cannot speak for any specific DNA repair mechanism. We agree with the reviewer that more clarification is needed to address this question, hence the following changes have been made to the main text.

New text: additions highlighted in yellow.

*The strandedness of annealing of CHEX-seq primers to ssDNA, apparent in the upstream of transcribed genes, suggests a scaffolding of proteins on one strand of the DNA (blocking CHEX-seq probe binding). This finding is intriguing given the dynamic back and forth between genes' double- and single-stranded status upon chemical perturbation, which emphasizes the role of chromatin 3D structure in transcriptional regulation. **The observed association between ssDNA and gene expression, DNA replication and maintenance indicates multiple possible sources of single-stranded chromatin, and also opens the door to intriguing questions such as how different single-stranded mechanisms coordinate in a living cell to produce and maintain the ssDNA landscape in homeostasis.** It is **also** tempting to hypothesize that the balance between the single and double-strand states of chromatin and the rate of interconversion **are** important to the degree of plasticity or vulnerability that any cell can exhibit.*

- Minor Concerns:
- 1. There are several typos throughout (for example ATAC-see instead of ATAC-seq), so some proofreading is necessary.

We apologize for this oversight. We have thoroughly proofread the manuscript and corrected the spelling and grammar issues. It is noteworthy that "ATAC-see" in fact is not a typo; it was published in an article entitled "ATAC-see reveals the accessible genome by transposase-mediated imaging and sequencing" (Chen et al. 2016).

- 2. In general, the methods are thorough and comprehensive. The authors provide GitHub repositories sharing their analysis pipeline and scripts.

Overview of CHEX analysis and experimental design: the technology works by using a degenerate sequence which anneals to ssgDNA. The degenerate sequence has a 5' barcode for single cell multiplexing and a 3' fluorescently tagged deoxynucleotide which can be activated with light (meaning the probe will function as a primer for cDNA synthesis when photoactivated). In this case, they use the 405nm laser on a confocal microscope for photoactivation. They then incubate the cells in the cDNA synthesis buffer and harvest individual cells manually with a glass micropipette.

Benchmarking in K562 cells: There seems to be a lot of priming sites indicated in the controls despite no CHEX-seq probes. This deserves some mention and explanation in the text. Presumably these "background" priming sites would not be barcoded and therefore could be subtracted? It is unclear just how the authors handle this background in the data analysis. In the methods they mention distinguishing CHEX priming from endogenous PCR priming using clipped 3' ends, but I failed to find specific mention of the non-CHEX priming sites in the priming counts section of the methods. This requires clarification in any revision of this otherwise excellent manuscript.

We thank for reviewer's thorough evaluation and constructive suggestions. In response we provide a brief reply to the reviewer's questions and indicate the consequent changes in the main text and figures. The 2-base clipped, partial barcode would in theory explain for $1/16 = 6.7\%$ background, but in general we found an excess of background (14.4% and 27.6% for single-cell and bulk samples, respectively). We analyzed the controls samples, and found several reasons for the background: (1) alignment artefacts which manifested as low sequence complexity in three Probe(+) Laser(-) controls; (2) problematic probe chemistry because the high-background

controls were almost all connected to the probe 517s; (3) C reads, namely, reads that contained the 3' primer ("pC") but not the 5' primer ("2p") and hence of lower quality than A or B reads, were found to be more prevalent in control samples, showing that the excess of background in controls were spurious.

Because of the general sparsity of CHEX-seq data in particular for single-cell samples, we have emphasized the sensitivity of the assay. Due to the background in "good" controls and the quality issues in "bad" controls, we did not directly mask or subtract the background from each non-control sample. We instead evaluated the impact of including the background for non-control samples. We did the following tests:

1) We compared the TSS coverages for (a) the control K562 samples, (b) the non-control K562 samples without background treatment, (c) the non-control samples with background subtracted (by masking the 2kb intervals of the priming sites pooling controls). We found that only (b) and (c) showed TSS peaks while for (a), neither Probe(+/-) Laser(-) nor mung bean digested K562 samples showed TSS peaks (revised Extended Data Fig 4C-E).

2) We compared K562 control and non-control samples in terms of the genomic distribution, and we observed significant deviation in the promoter and the distal intergenic regions (new Extended Data Fig 4F).

3) We compared K562 control and non-control samples in their enrichment in ENCODE ssDNA related epigenomes. Non-control K562, including or excluding the background, showed significantly stronger enrichment than control samples (new Extended Data Fig 4G, left).

4) We compared K562 control and non-control samples in their enrichment in K562 transcriptome (RNA-seq or GRO-seq). We observed ubiquitously higher fold-of-enrichment in non-control samples than in control samples, for all genic/sub-genic regions and for both RNA-seq and GRO-seq datasets (new Extended Data Fig 4G, right).

Therefore, we concluded that the background does not significantly impact our findings. We anticipate an improved signal-to-noise ratio in future experiments where we can increase the photo-activation efficiency with the purchase of a new laser system (enabling lower wavelength uncaging) which will increase sensitivity and further decreasing the background noise.

Below we list the changes in the main text.

New text:

After a series of QC filtering (for details see Methods: Data Preprocessing), the total number of priming sites in each non-control K562 sample varies from 305 to 60,437 (median = 2,640) for single cells, and from 30,118 to 85,382 (median = 53,357) for multi-cell (bulk) samples (Extended Data Fig 4). As a type of negative control, we treated 14 K562 cells with mung bean nuclease to digest ssDNA, and we observed significant reduction in the total number of priming sites, ranging from 1,139 to 4,632 (median = 1,890). significant reduction in the total number of priming sites, ranging from 1,139 to 4,632 (median = 1,890). The background is due to mung bean nuclease's incomplete digestion of ssDNA. Further, we carried out two other types of control experiment: (1) with barcoded probes but no laser activation ("Probe(+) Laser(-)"); (2) with neither barcoded probes nor laser activation ("Probe(-) Laser(-)"). Probe(-) Laser(-) controls showed greater reduction: only a median of 381 or 14,704 total priming sites observed in single cells or bulk samples, respectively (Extended Data Fig 4A, left), suggesting an overall false positive rate of 14.4% and 27.6% in single or bulk samples, which is higher than the expected 6.7% (1/16 for the 2bp-clipped primer induced endogenous priming). Several control samples were found to have reads of low sequence complexity due to alignment artefacts or are linked to a specific probe (517s) which could have problematic annealing or extension chemistry (for details see Methods:

Background Estimation and Processing). For other controls, we noted that C reads, i.e. reads with lesser barcode/primer quality (for definition see Methods: CHEX-seq ssDNA Calls and Priming Counts) were more prevalent in controls than non-controls (Extended Data Fig 4A, right), indicating that the background in these controls are spurious. Because CHEX-seq priming was sparse particularly in single cells, to ensure that we report as much of the complexity ssDNA regions as possible we prioritized the sensitivity and conducted a detailed examination of the background reads and showed that the impact of the background in non-control samples is negligible (for details see Methods: Background Estimation and Processing).

New text:

CHEX-seq ssDNA Calls and Priming Counts: Based on the presence and format of 5' end T7 barcode and 3' end pC primers, the alignments were classified into four categories: A, B, C, and D, among which A to C were divided into subclass 1 or 2, depending on the presence of read-through or not (Fig S3). For the sake of a balanced sensitivity and specificity, class A (both proper 2p and pC primer) and B reads (proper 2p/barcode primer only) were defined as good quality and merged as the AB reads. Then barcode locations were tracked in 20bp bins across the genome. For each sample with k cells, the number of reads whose barcodes hit the same bin were reduced to a maximum $2 \times k$, given the diploidy of the samples in this study. We prioritized the barcode/primer quality ($A1 > A2 > B1 > B2 > C1 > C2$) and the mapping quality (STAR alignment score, mapped length) when deciding the duplicate priming reads. We examined the data quality at each barcode/primer criterion threshold (A only, A and B merged, A, B and C merged) (Supplementary Information Table 3) and chose the A, B and C merged processing for the sake of maximal sensitivity given the sparsity of CHEX-seq priming. To evaluate the consequence of the inclusion of the C reads, we performed comprehensive analyses and showed that the impact of relaxed barcode/primer quality selection is negligible. (for details see Methods: Background Estimation and Processing). To count gene-level priming frequency, we defined various genic regions and sub-genic regions (Fig S8A), based on Ensembl gene models from R packages *EnsDb.Hsapiens.v86* (human) and *EnsDb.Mmusculus.v79* (mouse) (95). For CHEX-seq priming site annotation, we used R package *GenomicRanges* (96) and *ChIPseeker* (97). The GO enrichment analysis was done with R package *clusterProfiler* (98, 99).

Added text:

Background Estimation and Processing: To estimate the false positives we performed three types of control experiments in K562: (1) with barcoded probes but no laser activation ("Probe(+) Laser(-)"), (2) neither barcoded probes nor laser activation ("Probe(-) Laser(-)"), (3) with barcoded probes and laser activation but samples were mung bean digested ("MungBean"). Using the Probe(-) Laser(-) controls, we estimated the background in non-control samples as 14.4% and 27.6%, for single-cell or bulk K562 (Extended Data Fig 4A, left). MungBean controls exhibited a higher background, due to an incomplete digestion of the ssDNA. Only Probe(+) Laser(-) samples showed surprisingly more reads than non-controls. Further dissection of the read class revealed that the signals in control samples arose almost all (>90%) from C reads, i.e. the reads without the 5' barcoded primer (2p) but with the 3' primer (pC), whilst the fraction was much smaller for non-control samples (Extended Data Fig 4A, right). We also inspected the sequence complexity of aligned reads in Probe(+) Laser(-) samples and observed a much lower complexity (~0.5) as compared with the non-control samples (~0.68). As most of the low-complexity controls carried the probe 517s (Extended Data Fig 4B), we suspected that their unusually high background was something specific to this probe.

We performed a detailed evaluation of the impact of the background. First, we examined the TSS coverage and found that only non-control K562, including or excluding the background, showed TSS peaks as an aggregate, although the pattern became weakened for individual cells

(Extended Data Fig 4C-E). Consistent with the disappearance of TSS peaks, we found that compared with the non-controls, Probe(+/-) Laser(-) exhibited a 33.3% increase (from 60.1% to 80.6%) in distal intergenic reads and a 61.1% decrease (from 12.2% to 4.8%) in promoter reads (Extended Data Fig 4F). Next, we compared Probe(+/-) Laser(-) and non-control samples in their enrichment in ENCODE ssDNA related epigenomes. Non-control K562, including or excluding the background, showed significantly higher ($p = 0.083$ or 0.0096 , respectively) fold-of-enrichment than Probe(+/-) Laser(-) samples (Extended Data Fig 4G, left). Lastly, we compared Probe(+/-) Laser(-) and non-control samples in their enrichment in K562 transcriptome (RNA-seq or GRO-seq), and we observed ubiquitously stronger fold-of-enrichment in non-control samples than controls for all genic/sub-genic regions and for both RNA-seq and GRO-seq datasets (Extended Data Fig 4G, right).

- Overall, this is will undoubtedly be an impactful technology that will contribute to multiple discoveries in the biological sciences over the next decade or more. However, it is my sense that the manuscript lacks a genuine analysis of the current limitations of CHEX-seq. Where are the current deficiencies or limitations in the technology that hinder its applicability, and in what ways does it need to be improved and optimized? For example, they mention a surprising number of these background priming sites in controls that didn't even receive CHEX probes? Based on the numbers reported in the K562 cells between negative controls and CHEX-seq processed cells, it looks like the number of non-CHEX priming sites may account for as much as 17% of total priming sites (14,704 in bulk control samples compared to 85,382 in bulk CHEX samples). Looking at the medians they report in single cells, they may account for 28% (748 of 2,640) of priming sites in the single cell experiments. It is important to note that the authors state that this was what the data looked like after QC filtering. It would be beneficial if the research team would further explore this question and try to explain why there are so many non-CHEX priming sites and what are the implications of this for the quality of CHEX-seq data. On the same note, in one section they mentioned that they performed a modified analysis by adjusting the barcode and primer class criterion as well as the alignment criterion in order to reduce spurious reads. Can the authors calculate an estimation for the proportion of total reads that could be spurious? In spite of these questions, the manuscript has every likelihood of being very impactful to the life sciences and deserves publication.

Having addressed most of these comments in the previous point, we here briefly reply to the reviewer's questions and then show the revised text.

*First, the background in control samples is resultant of a combination of two major factors: (1) the preference of sensitivity has led to a particularly high number of false positive reads called in control samples. We have shown in Extended Data Fig 4 B-G that the background in non-control samples is unlikely to impact our major conclusions; (2) several control experiments did not appear to be good quality, in particular the Probe(+) Laser(-) controls. The background in non-control samples are estimated to be 14.4% and 27.6% in single-cell and bulk samples, respectively. For a complete account, please see the added **Methods: Background Estimation and Processing** section as shown in the previous point.*

*Second, we have expanded the **Discussion** section. We acknowledged the limitations of the current version of CHEX-seq. We also highlight future plans to address these limitations, such as using the optimal uncaging laser wavelength to increase the amount of probe activation, etc. The changed main text is as follows:*

New text:

Although the present data show the utility of CHEX-seq analysis in chromatin structure research, the current protocol can be optimized in several ways. First, in choosing to prioritize sensitivity of ssDNA region detection the background signal was higher than we would like. We detected a median of 2,640 ssDNA sites in single K562 cells, whilst single-cell ATAC-seq reported a median of 7,125 peaks in K562 (12). The difference may reflect biological differences in the ratio of dsDNA to ssDNA regions in the genome, but it also reflects a lower CHEX probe photo-activation yield due to the use of the 405nm uncaging laser line. The optimal 365nm wavelength was not available for these studies but should produce a 10X increase in uncaging efficiency and more activated CHEX-seq probe, which is expected to increase the number of detected ssDNA regions. CHEX-seq is ideal for analysis of single-strand chromatin in multimodal analyses and would benefit from decreasing the time it takes for the CHEX-seq *in situ* annealing and *in situ* transcription. This will be explored with optimized hybridization and reaction conditions that maintain the ability to multimodal analyze other chemicals in the same cell queried by CHEX-seq including proteins and metabolites. Further, while CHEX-seq works on immunostained cells in fixed tissue sections, to analyze the ssDNA chromatin landscape of thousands of *in situ* localized cells the approach needs to be transitioned into a high-throughput process. As CHEX-seq is a light activated process, to make it high throughput for single cells will require automated dynamic photomasking which will best be accomplished by developing AI algorithms to control this "on-scope" process. Despite these limitations, CHEX-seq revealed the genome-wide landscape of *in situ* single-stranded non-B-form DNA, and further established the association of ssDNA in transcribed genes (especially their intronic regions) with gene transcription. It also highlighted the association of ssDNA in distal intergenic areas with genome replication, repair and maintenance. Our data suggests an important regulatory role for the dynamic interconversion and ratio of single- to double-stranded DNA in chromatin dynamics. As such, the ability to perform single-cell, spatial open-chromatin, non-B-form DNA analysis on immuno-identified single neuronal cells promises to provide high resolution spatial genomic landscape analysis of dynamic circuit function and disease-associated dysfunction in functionally relevant cells.

Response to Reviewer 2's comments:

We again thank the reviewer for the constructive criticisms and suggestions.

- Lu et al described a new method, termed CHEX-seq, to detect single cell Genomic ssDNA with catalytical capacity. It is an interesting method, and the manuscript is well written and organized. Some of issues need to be addressed before it is published.

1, In the abstract and main text, they claimed they used formalin-fixed cells, actually it is 4% formaldehyde instead of formalin, which is different from formalin. The description has to be accurate. Otherwise, they need to prove: CHEX-seq works in formalin-fixed paraffin embedding clinical samples.

We have corrected the inaccuracy. The changes to the main text are as follows:

New text:

*Here we describe the CHEX-seq (CHromatin EXposed) assay that identifies single-stranded open chromatin *in situ* in individual, 4% paraformaldehyde fixed cells.*

- 2, In the workflow of CHEX-seq, No RNaseA or the other enzyme was used to digest native RNA in cells, would that be the issue for the potential contamination of CHEX-seq?

We thank the reviewer for being meticulous. Native RNA is not an issue for CHEX-seq as DNA Polymerase I is a DNA template specific polymerase.

- 3, When replication happens in the genome, ssDNA could be captured. Indeed, Figure 1 B, RepOrig is in the same branch of FAIRE, DNASE, ATAC. Could author compare CHEX-seq data with RepOrig data in genome wide to learn something?

Our data (Fig 1B) showed that RepOrig is relatively close to the four open-chromatin assays as a single branch, but we don't have more evidence to assert whether CHEX-seq is significantly closer to RepOrig than the other three assays. Judging by the genomic association test result (Supplementary Table S4), the quantile (0.067) of CHEX-seq's enrichment in RepOrig is two times higher than reported for the other assays (0.032, 0.032, 0.032 for ATAC-, DNase-, FAIRE-seq, respectively). To avoid over-interpretating the data, we report that CHEX-seq shows slightly more enrichment in RepOrig than other three assays.

- 4, Figs. 1 and 7: Author Showed that with a close comparison between the two assays (Fig. 1D and EX5A). CHEX-seq differs from ATAC-seq in having a wider peak with an extended slope of 5' to the TSS. It means that CHEX-seq captures additional information regarding transcriptional initiation events. In vitro, it has been shown that SSDNA is involved in catalyzing porphyrin metalation (Fig. 7). Could you clarify the evidence linking in vivo (Fig. 1) and in vitro (Fig. 7) events that CHEX-seq captures the SSDNA events other than DNA replication and associated events such as DNA methylation maintenance?

It is an intriguing question as to whether putative endogenous gDNAzyme sequences are correlate with any of the other functions of genomic DNA. We don't have evidence, nor did we intend to "link", ssDNA's transcription or replication/repair/maintenance functions in vivo, to its porphyrin metalation-related, enzymatic functions in vitro. Unlike transcription or replication/repair/maintenance, where there have been many RNA-seq, GRO-seq or ChIP-seq data against which we can test the correlation of CHEX-seq predicted ssDNA; endogenous DNAzymes are still much understudied and in vivo DNAzyme data are absent. We hope our data will contribute to expanding these research efforts.

- 5, Fig. 3: As it is mentioned in the K562 cells (Fig. 1C), the highest proportion of priming sites is found in the intergenic regions (> 60.1%), followed by introns (~26.1%), and then by promoter regions (less than 5kb to the TSS) (~12.2%). So, it means more than 86% of CHEX-seq priming sites are located in intergenic sites and non-coding parts of the genome other than the coding sequence. Does it mean that introns and intergenic sites of the genome are more likely to have it in the SSDNA state? Does CHEX-seq predict the most frequently occurring enzymatic activity by using these SSDNA originating from intergenic sites and introns?

We thank the reviewer for raising the interesting question. We don't believe that introns and intergenic sites are more likely to contain ssDNA. In Fig 1C we reported the number of ssDNA priming sites in each genomic region, however the data were not normalized to the different sizes of the genomic regions. If we calculate the priming sites per kilobase for exons, introns and intergenic regions that received CHEX-seq priming events, we observed (please see the figure below) small but significant differences in the length normalized ssDNA density in between the three groups—exons (median 0.177) > intergenic regions (median 0.116) > introns (median 0.0761).

Regarding DNase enzymatic activity, we do not have sufficient data to predict the rules governing which genomic regions tend to give rise to more enzymatic activity.

- Fig. 7: As you have mentioned, endogenous ssDNA have sequence similarity with manufactured DNases, and these sequences exhibit *in vitro* catalytic activity, but you found only three of them in the genome that have *in vitro* catalytic activity. What about the rest of the other sequences that failed to show enzymatic activity? Could you elaborate in detail? Could author provide supportive direct evidence for ssDNA catalyzing enzymatic activity *in vivo*?

New text:

Of the 2200 ssDNA regions predicted to contain gDNase (genomic DNase) sequences, a subset of those that correspond to DNases that metalate porphyrin were selected to screen for enzymatic activity. *Nine of the 95 identified CHEX-seq predicted porphyrin DNA metalation gDNase ssDNA regions were functionally tested in vitro, with 6 exhibiting catalytic activity and three showing no activity.*

At the moment, we cannot provide direct evidence for in vivo ssDNA catalytic activity of the in vitro positive porphyrin metalation gDNases. Normally, one would perform a combination of knockout, mutational and addback experiments to show in vivo functionality but with the large number of regions showing sequence and structure-constrained porphyrin metalation activity, it is difficult to utilize the standard approaches to elaborate on the in vivo functionality. However, the discovery that sequences within the genome have the capacity to catalyze reactions in vitro, will undoubtedly stimulate research in assessing the in vivo functionality of sequences predicted to metalate porphyrin as well as other predicted catalytic activities.

REVIEWERS' COMMENTS

Reviewer #1 (Remarks to the Author):

The authors have more than adequately addressed the concerns I raised in the original review of this interesting manuscript. I believe I have no further issues and support its publication.

Reviewer #2 (Remarks to the Author):

Regarding CHEX-seq predicting single-cell genomic ssDNA state and its functional value. As the author claimed "It is an intriguing question as to whether putative endogenous gDNAzyme sequences are correlate with any of the other functions of genomic DNA. We don't have evidence, nor did we intend to "link", ssDNA's transcription or replication/repair/maintenance functions in vivo, to its porphyrin metalation-related, enzymatic functions in vitro.....".

And also claimed that they needed additional experiments to justify its in vivo function "At the moment, we cannot provide direct evidence for in vivo ssDNA catalytic activity of the in vitro positive porphyrin metalation gDNAzymes. Normally, one would perform a combination of knockout, mutational and addback experiments to show in vivo functionality but with the large number of regions showing sequence and structure-constrained porphyrin metalation activity, it is difficult to utilize the standard approaches to elaborate on the in vivo functionality".

It seems that the title of this article is confusing to the audience; it doesn't provide evidence in terms of in vivo functionality of ssDNA predicted by CHEX-seq. To be clearer, I suggest some modification in the title for publication as "CHEX-seq Detects Single-Cell Genomic ssDNA".

Response to Reviewers

One reviewer had no suggested changes while the other reviewer only suggested a title change. We have changed the title to clear up any misconceptions about what ssDNA means and would like to use the following as the title, as we believe it well reflects the work that is presented in the manuscript.

CHEX-seq Detects Single-Cell Genomic Single-Stranded DNA With Catalytical Potential